# Curse of High Dimensionality Issue in Transformer for Long-context Modeling

**Shuhai Zhang** [* 1 2]  **Zeng You** [* 1 3]  **Yaofo Chen** [1]  **Zhiquan Wen** [1]  **Qianyue Wang** [1]  **Zhijie Qiu** [1]
**Yuanqing Li** [1 2]  **Mingkui Tan** [1 4]

## Abstract

Transformer-based large language models (LLMs) excel in natural language processing tasks by capturing long-range dependencies through self-attention mechanisms. However, long-context modeling faces significant computational inefficiencies due to *redundant* attention computations: while attention weights are often *sparse*, all tokens consume *equal* computational resources. In this paper, we reformulate traditional probabilistic sequence modeling as a *supervised learning task*, enabling the separation of relevant and irrelevant tokens and providing a clearer understanding of redundancy. Based on this reformulation, we theoretically analyze attention sparsity, revealing that only a few tokens significantly contribute to predictions. Building on this, we formulate attention optimization as a linear coding problem and propose a *group coding strategy*, theoretically showing its ability to improve robustness against random noise and enhance learning efficiency. Motivated by this, we propose *Dynamic Group Attention* (DGA), which leverages the group coding to explicitly reduce redundancy by aggregating less important tokens during attention computation. Empirical results show that our DGA significantly reduces computational costs while maintaining competitive performance. Code is available at https://github.com/bolixinyu/DynamicGroupAttention.

## 1. Introduction

Transformer-based large language models (LLMs) (Touvron et al., 2023a;b) have advanced in various natural language processing tasks (Brown et al., 2020; Wei et al., 2022), ex-

hibiting emergent abilities like few-shot learning and complex reasoning (Schaeffer et al., 2023). These capabilities stem primarily from their ability to model long-range dependencies through self-attention (Vaswani, 2017; Schaeffer et al., 2023). However, a key challenge in long-context modeling is the redundancy in attention computation: while attention weights are often *sparse*, meaning many tokens contribute minimally to predictions, all tokens still consume *equal* computational resources. This substantial inefficiency raises the question: how can we reduce *redundant* computations without sacrificing model performance?

Existing approaches (Xiao et al., 2024; Han et al., 2023; Chen et al., 2024) often tackle this issue by discarding some tokens to simplify attention computations and reduce costs. However, while effective in specific scenarios, such methods risk disrupting token interactions, especially in tasks requiring comprehensive context understanding, *e.g.*, question answering (Lu et al., 2022; Singhal et al., 2025) and document summarization (Cao et al., 2017; Pasunuru et al., 2021). Token removal may lead to incomplete or inaccurate context comprehension, impairing model performance. Thus, the key challenge is how to minimize redundant attention computations while maintaining critical token interactions.

Traditional probabilistic sequence modeling, such as autoregressive models (Vaswani, 2017; Huang et al., 2018; OpenAI, 2023), treats long-context redundancy implicitly by processing the entire context as a block during next-token prediction, making it difficult to analyze the above issue. To better understand and analyze redundancy, we reformulate the probabilistic sequence modeling as a **supervised learning task** in Section 3. This reformulation enables us to separate relevant and irrelevant tokens and provides a clearer view of redundancy, inspiring us to develop more efficient methods for long-context modeling.

To gain a deeper understanding of the redundancy in a long context, we provide a theoretical analysis of the sparsity of attention weights in transformers in Section 4. Our analysis shows that only a small subset of tokens significantly contributes to the target representation, while many tokens offer little value to overall model performance, leading to inefficient use of computational resources. To better understand this issue and inspire new attention, we formulate

---

[*]Equal contribution  [1]South China University of Technology  [2]Pazhou Laboratory  [3]Peng Cheng Laboratory  [4]Key Laboratory of Big Data and Intelligent Robot, Ministry of Education. Correspondence to: Mingkui Tan <mingkuitan@scut.edu.cn>.

*Proceedings of the 42$^{st}$ International Conference on Machine Learning*, Vancouver, Canada. PMLR 267, 2025. Copyright 2025 by the author(s).

attention optimization as a linear coding problem (Ryan & Lin, 2009). In long-context modeling, the sparsity of attention weights often causes learning unstable and inefficient (Lounici et al., 2011; Huang & Zhang, 2010). To address this, we propose a **group coding strategy**, which aggregates tokens into meaningful groups. Our theoretical analysis demonstrates that the group coding is more robust to random noise and offers a more stable and efficient learning process. This suggests that grouping mechanisms can effectively reduce redundancy in long-context modeling, offering a new perspective on transformer optimization.

To further exploit the group mechanism to reduce computational redundancy in LLMs, we propose a **Dynamic Group Attention (DGA)** mechanism, which explicitly reduces redundancy in attention computation without sacrificing critical token interactions, as detailed in Section 5. The core idea of DGA lies in dynamically grouping and aggregating less important tokens during attention computation, as illustrated in Figure 1 and Algorithm 1. Specifically, 1) DGA identifies tokens that contribute minimally to the attention process, grouping them together and aggregating their representations before performing the attention operation. This reduces the number of individual tokens involved in attention calculations, thereby significantly reducing computational complexity. 2) Crucially, the more important tokens, which are vital for maintaining the model's performance, are handled individually, ensuring their interactions remain preserved and accurately represented. 3) Furthermore, DGA introduces complementary keys and values for tokens restricted from accessing group information due to the autoregressive nature of LLMs. By focusing the attention mechanism on the most relevant tokens and reducing the redundancy of less informative ones, DGA not only reduces redundancy computation but also maintains essential token interactions. Empirical results demonstrate the superiority of our DGA for long-context modeling.

We summarize our main contributions as follows:

- A supervised reformulation of sequence modeling for redundancy analysis: We reformulate the sequence modeling as a supervised learning task, enabling the separation of relevant and irrelevant tokens in long-context modeling. It provides a clearer understanding of redundancy and inspires more efficient methods for long-context modeling.

- Theoretical analyses of attention sparsity and group coding strategy: We theoretically analyze the sparsity of attention weights in transformers, showing that only a small subset of tokens significantly contributes to the target representation. To inspire new attention, we formulate attention optimization as a linear coding problem. Based on this, we propose a group coding strategy and theoretically show that the group coding is more robust to random noise

and offers a more stable and efficient learning process.

- Dynamic group attention for long-context modeling: We propose Dynamic Group Attention (DGA) to address redundancy in attention computation for long-context modeling. DGA dynamically aggregates less important tokens into meaningful groups, significantly reducing computational costs while preserving critical token interactions. Empirical results on diverse long-context tasks demonstrate that our DGA significantly reduces computation while maintaining competitive performance.

## 2. Related Work

**Efficient transformer.** Early efforts focus on introducing sparsity into attention mechanisms. For instance, Reformer (Kitaev et al., 2020) uses Locality Sensitive Hashing (LSH) to group similar tokens into buckets, reducing computational demands. Similarly, Longformer (Beltagy et al., 2020) combines global attention for key tokens with local sliding window attention to handle longer texts. Big Bird (Zaheer et al., 2020) further integrates global, local, and random attention strategies to capture both long-range and local contexts.

Another line of research approximates attention mechanisms to achieve linear complexity. Performer (Choromanski et al., 2021) introduces a kernel-based approximation of softmax attention, reducing memory and computational costs. Similarly, Linear Transformer (Katharopoulos et al., 2020) and RetNet (Sun et al., 2023) reformulate self-attention as a linear dot-product of kernel feature maps. Additionally, Sun et al. (2021) learn parameterized hash functions for queries and keys to enhance efficiency. HyperAttention (Han et al., 2024) refines attention approximation by measuring problem hardness with fine-grained parameters.

Recent works further advance long-context modeling. LongLoRA (Chen et al., 2024) partitions tokens into groups and shifts group partitions to facilitate inter-group communication for modeling efficiency. StreamingLLM (Xiao et al., 2024) and LM-Infinite (Han et al., 2023) prioritize attention on the initial and final tokens, ignoring intermediate tokens to optimize attention. KVMerger (Wang et al., 2024b) focuses on KV cache compression via Gaussian-kernel-based Key clustering but ignores attention computation redundancy. Some methods focus on dynamically adapting sparsity patterns. For instance, MInference (Jiang et al., 2024) identifies three unique sparse attention patterns and dynamically applies them during inference.

Despite these advancements, such methods often rely on fixed sparsity patterns, which may sacrifice important token interactions. This can degrade performance in tasks requiring fine-grained interactions. Recently, CCA-Attention (Chen et al., 2025) leverages grouped aggregation of intra-token interactions combined with local sliding windows for

efficient attention. In contrast, our method dynamically identifies critical tokens and selectively aggregates only unimportant tokens while preserving essential interactions via complementary tokens. This ensures adaptive redundancy reduction without sacrificing key contextual dependencies.

**Long-context modeling.** Extending the context window of LLMs to handle long sequences is critical for tasks requiring deep understanding. Recently, A plethora of work has attempted to extend the context length of LLMs (Chen et al., 2023; Yen et al., 2024; Mohtashami & Jaggi, 2023; An et al., 2024; Tworkowski et al., 2023). For instance, Position Interpolation (Chen et al., 2023) linearly downscales input position indices to fit within the original context window size, enabling RoPE-based LLMs to handle longer sequences. Yen et al. (2024) introduce a small encoder to process long inputs in chunks, allowing a frozen decoder to cross-attend to additional contexts, thus extending the context length without modifying the core architecture. In contrast, An et al. (2024) propose Dual Chunk Attention, enabling models like Llama2-70B to support context windows exceeding 100,000 tokens without extra training, by decomposing long-sequence attention into chunk-based modules that capture both intra-chunk and inter-chunk dependencies.

Other approaches focus on modifying position embeddings to extend context length, such as positional skipping (Zhu et al., 2024), Yarn (Peng et al., 2024), and RoPE extrapolation (Liu et al., 2024b). While these methods primarily address position embedding limitations, our approach is orthogonal, focusing on reducing computational redundancy through dynamic token grouping and aggregation.

# 3. Sequence Modeling to Supervised Learning

## 3.1. Traditional Probabilistic Sequence Modeling

Traditional probabilistic sequence modeling (Vaswani, 2017; Wang et al., 2024a) in language models typically involves generating a token sequence by predicting each next token based on previously generated ones. Given a token sequence $\{\mathbf{x}_1, \mathbf{x}_2, \ldots, \mathbf{x}_L\}$, the training objective for a model $\theta$ is to maximize the likelihood of the entire sequence:

$$\max_{\theta} \prod_{i=1}^{L} P_{\theta}(\mathbf{x}_i | \mathbf{x}_1, \mathbf{x}_2, \ldots, \mathbf{x}_{i-1}). \tag{1}$$

During the generation, the model selects the most likely token at each step via $\mathbf{x}_i = \arg\max_{\mathbf{x}} P(\mathbf{x} | \mathbf{x}_{<i})$. This autoregressive process iteratively applies the same mechanism until the desired sequence length is reached.

**Limitations of probabilistic sequence modeling for analysis.** While this approach ensures coherent and contextually relevant token generation, it struggles with long sequences. As the sequence length increases, the model often encoun-

ters more irrelevant information, leading to inefficiencies in both computation and optimization. Traditional sequence modeling treats the entire context as a monolithic block due to its conditional modeling paradigm, making it difficult to identify or eliminate redundant tokens. Moreover, the implicit handling of redundancy in traditional sequence modeling (*e.g.*, self-attention (Vaswani, 2017)) limits the ability to optimize computational resources effectively.

## 3.2. Supervised Reformulation for Sequence Modeling

Recall that *next-token prediction* (Zhang et al., 2024) in traditional sequence modeling is $\mathbf{x}_i = \arg\max_{\mathbf{x}} P(\mathbf{x}|\mathbf{x}_{<i})$. We can reformulate it as $\mathbf{y} = \arg\max_{\mathbf{y}} P(\mathbf{y}|C(\mathbf{y}))$ with $C(\mathbf{y}) = \{\mathbf{x}_1, \mathbf{x}_2, \ldots, \mathbf{x}_{i-1}\}$ representing the context preceding the target token $\mathbf{y}$. This can be interpreted as predicting label $\mathbf{y}$ based on the input $C(\mathbf{y})$, resembling the supervised learning mechanism (Hastie et al., 2009).

Building on this insight, we formalize probabilistic sequence modeling as a **supervised learning task**. Given a training corpus $\{(C(\mathbf{y}_i), \mathbf{y}_i)\}_{i=1}^{n}$, where $\mathbf{y}_i \in \mathcal{V}$ is a token from the vocabulary $\mathcal{V}$ and $C(\mathbf{y}_i)$ is its context, the task is to maximize the likelihood of predicting conditioned on all $C(\mathbf{y}_i)$:

$$\max_{\theta} \prod_{\mathbf{y} \in \mathcal{V}, C(\mathbf{y})} P_{\theta}(\mathbf{y}|C(\mathbf{y})). \tag{2}$$

Denoting $f$ as the feature extractor in the last layer of the LLM, the probability $P_{\theta}(\mathbf{y}|C(\mathbf{y}))$ can be reformulated as

$$P_{\theta}(\mathbf{y}|C(\mathbf{y})) = \frac{\exp(\mathbf{z}_k)}{\sum_{j=1}^{K} \exp(\mathbf{z}_j)}, \tag{3}$$

where $k$ is the index of the token in the vocabulary, and $\mathbf{z} = \mathbf{w}^{\top} f(C(\mathbf{y})) \in \mathbb{R}^K$ represents the logits of $\mathbf{y}$. Here, $\mathbf{w} \in \mathbb{R}^{d \times K}$ is the weight matrix of the final fully connected layer, and $K = |\mathcal{V}|$ is the vocabulary size. This is a standard supervised learning paradigm (Hastie et al., 2009; Nasteski, 2017). Further discussion on the relations between sequence modeling and supervised learning is in Appendix B.

**Remark 1.** *1) Ideally, for each target token* $\mathbf{y}$*, it is crucial to collect all relevant contexts* $C(\mathbf{y})$ *for accurate prediction. 2) To train a promising model, it is essential to gather the comprehensive context associated with each* $\mathbf{y}$*. This can be achieved by collecting longer corpora (Fu et al., 2024). However, longer contexts inevitably introduce redundant tokens, leading to unnecessary computational overhead and optimization challenges (Altman & Krzywinski, 2018).*

**Advantages of supervised sequence modeling.** The supervised sequence modeling provides a more structured approach to understanding long-context redundancy, although they are theoretically equivalent. It allows us to separate the context $C(\mathbf{y})$ into relevant and irrelevant parts, allowing a more focused examination of the redundancy issue. By

casting the problem as supervised learning, we can precisely determine which tokens in $C(\mathbf{y})$ are vital for predicting $\mathbf{y}$, and which tokens may be discarded or aggregated.

**Redundancy in long contexts.** From the above view, the context $C(\mathbf{y})$ with redundancy can be formalized as

$$C(\mathbf{y}) = \{\mathbf{x}_1^{\mathrm{R}}, \mathbf{x}_2^{\mathrm{IR}}, \mathbf{x}_3^{\mathrm{R}}, \mathbf{x}_4^{\mathrm{IR}}, \mathbf{x}_5^{\mathrm{IR}}, \ldots, \mathbf{x}_{L-1}^{\mathrm{R}}, \mathbf{x}_L^{\mathrm{IR}}\}, \quad (4)$$

where $\mathbf{x}^{\mathrm{R}}$ denotes relevant tokens that contribute to predicting the target token $\mathbf{x}$, and $\mathbf{x}^{\mathrm{IR}}$ denotes irrelevant tokens that introduce unnecessary noise or computation.

**Challenges.** The redundancy becomes particularly pronounced in transformers, which process all tokens with equal computational cost. However, not all tokens contribute equally to predicting $\mathbf{y}$: many tokens in $C(\mathbf{y})$ offer negligible or redundant information, inflating computational requirements and impairing optimization efficiency. This motivates us to analyze the redundancy within $C(\mathbf{y})$ and find more effective methods for long-context modeling.

## 4. Redundancy in Self-attention: From Mechanism to Coding Perspective

Section 3 reformulates long-context modeling as a supervised learning task (Eqn. (2)), which explicitly separates *relevant* (critical for predictions) and *irrelevant* (redundant for context) tokens. This formulation provides a structured foundation to investigate how redundancy manifests in attention computations and motivates our theoretical exploration from both the mechanism and coding perspectives.

### 4.1. Redundancy in Self-attention Mechanism

**Self-attention in LLMs.** Self-attention (Vaswani, 2017) is a key mechanism in LLMs that allows the model to assign the importance of different tokens for long-context modeling. Formally, given input $\mathbf{X} \in \mathbb{R}^{L \times d}$, self-attention is defined as:

$$\mathbf{Att} = \mathrm{softmax}\left(\frac{\mathbf{Q}\mathbf{K}^\top}{\sqrt{d}}\right)\mathbf{V}, \quad (5)$$

where $\mathbf{Q}=\mathbf{X}\mathbf{W}^Q$, $\mathbf{K}=\mathbf{X}\mathbf{W}^K$, and $\mathbf{V}=\mathbf{X}\mathbf{W}^V$ represent the query, key, and value matrices, respectively. Here, $\mathbf{W}^Q, \mathbf{W}^K, \mathbf{W}^V \in \mathbb{R}^{d \times d}$ are learnable projection weights and $d$ is the dimensionality of the token embeddings. For simplicity, we denote the attention weight as $\mathbf{A}=\mathrm{softmax}(\mathbf{Q}\mathbf{K}^\top/\sqrt{d})$, which captures the pairwise relevance between tokens, determining how much influence of each token on the representation of the other tokens. To facilitate the understanding of the attention mechanism, we rewrite the attention of $i$-th token separately:

$$\mathbf{Att}_i = \sum_{j=1}^{L} \mathbf{A}_{i,j}\mathbf{V}_j \in \mathbb{R}^{1 \times d}, \ i = 1, \ldots, L, \quad (6)$$

where $\mathbf{A}_i=\mathrm{softmax}(\mathbf{Q}_i\mathbf{K}^\top/\sqrt{d})$. This implies that the attention of a token is computed as a linear combination of the values of other tokens, weighted by the importance obtained from the dot product of its query and the keys of other tokens. This mechanism enables the model to consider long-range dependencies for long-context modeling.

**Redundancy in attention weights.** To analyze the inherent redundancy in long-context modeling within LLMs, we focus on the representation of the context $C(\mathbf{y})$ in a simplified model: a single-layer transformer with single-head attention. Given a sequence $C(\mathbf{y}) \in \mathbb{R}^{L \times d}$, denoted as $\mathbf{X}=[\mathbf{x}_1; \ldots; \mathbf{x}_L]$ for simplicity, the representation of the last token $f(\mathbf{X})$ can be formulated as:

$$f(\mathbf{X}) = \alpha_1 \mathbf{V}_1 + \alpha_2 \mathbf{V}_2 + \cdots + \alpha_j \mathbf{V}_j + \cdots + \alpha_L \mathbf{V}_L, \quad (7)$$

where $\alpha_j = \frac{\exp(\mathbf{Q}_L \cdot \mathbf{K}_j)}{\sum_{l=1}^{L} \exp(\mathbf{Q}_L \cdot \mathbf{K}_l)}$ represent the $j$-th weight.

The attention weights $\alpha_j$ determine the contribution of each token's value vector $\mathbf{V}_j$ to the target representation $f(\mathbf{X})$. However, in long sequences, a significant portion of the tokens in the context are irrelevant to the target representation as in Eqn. (4), resulting in sparse and concentrated attention distributions, *i.e.*, a few weights $\alpha_j$ are significantly greater than most weights. To facilitate understanding, we analyze the sparsity of attention weights in the following:

**Theorem 1.** *(Sparsity on attention weights) Consider $\rho \in (1/L, 1]$ as a sparse rate, we say that the weight $\alpha$ is $\rho$-sparse when there exists at least one probability greater than $1/(L\rho)$. Let $\xi = \mathbf{K}\mathbf{Q}_i \in \mathbb{R}^L$, then the probability of $\alpha$ being $\rho$-sparse $P_{sparse}(L, \rho)$ is given by*

$$P_{sparse}(L, \rho) \geq \max_{x>0} 1 - [P_{head}P_{tail}]^L, \quad (8)$$

*where $P_{head} = P\{\exp(\xi_j) \leq x\}$ with $j$ being some index of $\alpha$ and $P_{tail} = P\{(L\rho - 1)x \leq \sum_{k \neq j}^{L} \exp(\xi_k)\}$.*

**Remark 2.** *Given $L$ and $\rho$, the probability $P_{sparse}(L, \rho)$ is influenced by the interplay between $P_{head}$ and $P_{tail}$ that are functions of $x$. A larger $x$ increases $P_{head}$ but decreases $P_{tail}$, leading to a balance where $P_{head}P_{tail}<1$. For sufficiently large $L$, the exponential amplification of $L$ suppresses $P_{head}P_{tail}$ further, ensuring a large $P_{sparse}(L, \rho)$.*

**Remark 3.** *$P_{sparse}(L, \rho)$ rigorously quantifies how sparsity intensifies with increasing context length $L$. It reflects a model's inherent ability to prioritize critical tokens across diverse contexts, e.g., $P_{sparse}(L, \rho)$ evaluated on xxx model rises sharply for $\rho=0.01$ as $L$ grows (see Figure 2(b)). Crucially, $\rho$-sparsity can be aggregated over multiple sequences to compute the average sparsity level (see Section 6.2), providing a model-level characterization of sparsity that generalizes beyond individual instances. This metric serves as a unified approach to evaluate and compare the efficiency of attention mechanisms in long-context modeling.*

The sparsity of $\alpha$ implies that only a few tokens in the context contribute meaningfully to the target representation. While this sparsity aligns with the nature of self-attention, the redundancy in the input context still increases the computational burden during training, as all tokens are processed equally regardless of their relevance. This not only increases training time but also impairs optimization efficiency.

### 4.2. Redundancy Analysis from Coding Perspective

In this work, we aim to seek an effective mechanism to alleviate the issue of attention redundancy for long-context modeling. Inspired by the form in Eqn. (7), we simplify the optimization process of a transformer into a linear coding problem (Ryan & Lin, 2009). This enables us to investigate the sparsity and redundancy of attention weights and inspires us to develop more efficient optimization approaches. Specifically, we will show how grouping mechanisms can reduce redundancy while improving optimization efficiency.

**Problem 1.** *(Linear coding problem for self-attention)*

$$\min_{\alpha} \left\| \sum_{j=1}^{L} \alpha_j \mathbf{V}_j - \mathbf{y} \right\|_2^2, \ s.t., \sum_{j=1}^{L} \alpha_j = 1, \ \alpha_j > 0. \quad (9)$$

*where $\mathbf{y}$ is the embedding of the target token.*

In this formulation, $\alpha_j$ represents the contribution of the $j$-th token's value vector $\mathbf{V}_j$ to the target representation, and its constraints are attributed to the softmax in the attention. Here, we consider each embedding $\mathbf{y}$ of the target token in the vocabulary and the parameters of $\mathbf{V}_j$ as fixed, simplifying the analysis of transformer optimization using linear coding techniques (MacKay, 2003; Ryan & Lin, 2009).

Due to the sparsity property established in Theorem 1, the attention weights $\alpha$ focus on a few key tokens, rendering most of the context redundant. However, this redundancy challenges optimization: *while sparse weights are beneficial, achieving stable and efficient learning is often hindered by noise sensitivity and overfitting to irrelevant tokens* (Lounici et al., 2011; Huang & Zhang, 2010).

One naive approach is to add some penalties during optimization. Unfortunately, each context $C(\mathbf{x})$ has its optimal weight within the whole tokens of the context. To address this, benefiting from the group sparsity (Huang & Zhang, 2010; Lounici et al., 2011), we propose a **group coding** strategy, reducing the inherent redundancy by partitioning the attention weights $\alpha$ into $k$ groups, *i.e.*, $\bar{\alpha} \in \mathbb{R}^k$. Each group *shares* a common weight, effectively pooling the contributions of grouped tokens. Let $G_1, \ldots, G_k$ denote the index sets from the token indices $\mathcal{I} = \{1, \ldots, L\}$, where $\mathcal{I} = \bigcup_{g=1}^{k} G_g$ and $G_i \bigcap G_j = \emptyset$ for $\forall i \neq j$. Formally, we define the group weight as $\bar{\alpha}_g = \frac{1}{|G_g|} \sum_{i \in G_g} \alpha_i$. The optimization problem is then reformulated as:

**Problem 2.** *(Group coding problem for self-attention)*

$$\min_{\bar{\alpha}} \left\| \sum_{g=1}^{k} \frac{\bar{\alpha}_g}{|G_g|} \sum_{j \in G_g} \mathbf{V}_j - \mathbf{y} \right\|_2^2, \ s.t., \sum_{g=1}^{k} \bar{\alpha}_g |G_g| = 1, \ \alpha_g > 0. \quad (10)$$

This group mechanism not only reduces the dimensionality of optimization but also reduces the redundancy observed in the attention. Aggregating tokens into meaningful groups can effectively compress redundant information and exploit the shared structure within the context. We summarize the advantages of this approach as follows:

1) **Improved Robustness to Noise**: The grouping mechanism smooths out the impact of random noise by averaging over multiple tokens in the same group. Mathematically, as shown in Theorem 2, the variance of weight changes is reduced by a factor of $1/m^2$, where $m$ is the group size.

**Theorem 2.** *Assume the weights before normalization are $\tilde{\alpha}$ obtained by the product of the query and key matrix, and $\alpha_j = \frac{\exp(\tilde{\alpha}_j)}{\sum_{l=1}^{L} \exp(\tilde{\alpha}_l)}$. Consider a Gaussian noise $\Delta\tilde{\alpha} \sim \mathcal{N}(\mathbf{0}, \sigma^2 \mathbf{I}_L)$ being added into $\tilde{\alpha}$ and each group size being $|G_g| = m$, the variance of the weight changes obtained via group coding in Eqn. (2) can be reduced by $1/m^2$.*

2) **Accelerated Optimization**: Most weights in long-context modeling are relatively small (see Theorem 1), making most gradient contributions also small; The group optimization reduces both the optimization dimension and gradient sparsity while enhancing stability. Theorem 3 shows that the group coding reduces the condition number (Edelman, 1988) of the objective function, enhancing convergence of the optimization, as verified in Section 6.3.

**Theorem 3.** *Denote $H$ and $\bar{H}$ as the Hessian matrix of the optimizations in (1) and (2), respectively. Let $\kappa(H) = \lambda_{\max}(H)/\lambda_{\min}(H)$ be the condition number (Edelman, 1988) of $H$, where $\lambda_{\max}$ and $\lambda_{\min}$ are the maximum and minimum eigenvalues of $H$, respectively. Consider each group size $|G_g| = m$, $\lambda_{\min}(H) > 0$ and $\lambda_{\min}(\bar{H}) > 0$, we have*

$$\kappa(\bar{H}) \leq \kappa(H). \quad (11)$$

Through the lens of the group linear coding, we highlight the redundancy in the long context can be mitigated by structured grouping. This approach not only reduces the computational overhead but also enhances model robustness and optimization efficiency, offering a feasible way to manage sparsity and redundancy in transformer optimization.

## 5. Dynamic Group Attention Mechanism for Long-context Modeling

Theorems 2 and 3 show the robustness and optimization efficiency of group coding in reducing redundancy, inspiring

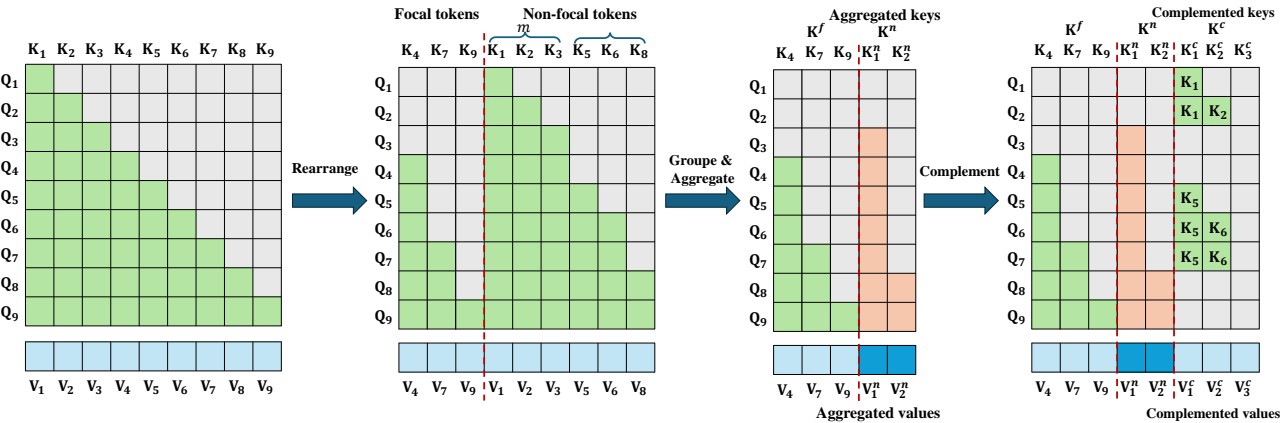

*Figure 1.* Overview of the proposed DGA attention. The DGA attention dynamically adjusts the computation based on token importance. First, DGA moves the key-value (KV) pairs of important tokens to the front (denoted as $\mathbf{K}^f$ and $\mathbf{V}^f$). For less important tokens, DGA groups and aggregates their KV pairs with a group size $m$ (denoted as $\mathbf{K}^n$ and $\mathbf{V}^n$). Finally, DGA introduces complementary KV pairs (denoted as $\mathbf{K}^c$ and $\mathbf{V}^c$) to enable access to group information for tokens restricted by the autoregressive nature.

---

**Algorithm 1** The pipeline of dynamic group attention.

---

**input** Matrices $\mathbf{Q}, \mathbf{K}, \mathbf{V} \in \mathbb{R}^{L \times d}$, group size $m$, importance rate $\gamma$.
**output** $\mathbf{Att} \in \mathbb{R}^{L \times d}$.
  1: Compute the importance score $s$ via Eqn. (16) and (17).
  2: Divide the tokens into *focal tokens* $\mathcal{T}^{\text{foc}}$ and *non-focal tokens* $\mathcal{T}^{\text{non}}$ based on $s$ and $\gamma$.
  3: Construct $\mathbf{K}'$ and $\mathbf{V}' \in \mathbb{R}^{L \times d}$ by moving the items of $\mathbf{K}$ and $\mathbf{V}$ corresponding to $\mathcal{T}^{\text{foc}}$ to the front positions.
  4: Construct $\mathbf{K}^{\text{group}}$ and $\mathbf{V}^{\text{group}}$ via Eqn. (12), (14), (15).
  5: Construct the causing mask $\mathbf{M}$ via Algorithm 2.
  6: **for** $i = 1, \ldots, L$ **do**
  7:   // Compute the attention for the query $\mathbf{Q}_i$.
  8:   $\mathbf{Att}_i = \text{Attention}^{\text{flash}}(\mathbf{Q}_i, \mathbf{K}^{\text{group}}, \mathbf{V}^{\text{group}}, \mathbf{M}_i)$.
  9: **end for**

---

us to explore a novel approach to address the computational inefficiencies in long-context modeling. Typical methods often use sparse mechanisms (Xiao et al., 2024; Han et al., 2023; Chen et al., 2024), discarding some tokens to simplify attention computation and reduce costs. However, such approaches risk disrupting the flow of information, particularly in tasks requiring comprehensive context understanding, such as question answering or document summarization.

**Overview of dynamic group attention.** To address the above issues, we leverage the advantages of the group mechanism and propose **Dynamic Group Attention (DGA)**, which explicitly reduces redundancy in attention computation without sacrificing essential token interactions. Specifically, we divide tokens into two parts based on the *importance score*: a small subset of significant tokens as **focal tokens**, while the less critical tokens are treated as **non-focal tokens**. Non-focal tokens are then grouped and aggregated, allowing the model to focus on aggregated representations

rather than individual tokens. The overview of our DGA is shown in Figure 1 and the algorithm is in Algorithm 1.

Formally, we give the grouped $\mathbf{K}^{\text{group}}$ and $\mathbf{V}^{\text{group}}$ as:

$$\mathbf{K}^{\text{group}} = \left[\mathbf{K}^f; \mathbf{K}^n; \mathbf{K}^c\right], \quad \mathbf{V}^{\text{group}} = \left[\mathbf{V}^f; \mathbf{V}^n; \mathbf{V}^c\right], \quad (12)$$

where $\mathbf{K}^f$ and $\mathbf{V}^f$ are the key and value of the focal tokens, $\mathbf{K}^n$ and $\mathbf{V}^n$ are the key and value of the non-focal tokens that are grouped and aggregated, $\mathbf{K}^c$ and $\mathbf{V}^c$ are the key and value complementing tokens that cannot access the group information due to the autoregressive nature. The detailed formulations are given in Eqn. (14) and Eqn. (15). Recall the definition of attention in Eqn. (6), we compute the attention with causing mask as:

$$\mathbf{Att}_i = \sum_j (\mathbf{A}_{i,j}^{\text{group}} \odot \mathbf{M}_{i,j}) \mathbf{V}_j^{\text{group}} \in \mathbb{R}^{1 \times d}, \quad (13)$$

where $\mathbf{A}_i^{\text{group}} = \text{softmax}\left(\mathbf{Q}_i \mathbf{K}^{\text{group}\top} / \sqrt{d}\right)$, $\mathbf{M}$ is the causing mask and $\odot$ is the element-product. We can implement this by the flash attention (Dao et al., 2022), simply formulated as $\text{Attention}^{\text{flash}}(\mathbf{Q}_i, \mathbf{K}^{\text{group}}, \mathbf{V}^{\text{group}}, \mathbf{M}_i)$.

### 5.1. Grouping for Self-attention

Given a sequence with $L$ tokens, we first employ an importance statistic to derive the importance score $s \in \mathbb{R}^L$ (see Eqn. (16) in Section 5.2) and divide all tokens into *focal tokens* (top-$\gamma$ important ones) and *non-focal tokens*, denoted as $\mathcal{T}^{\text{foc}}$ and $\mathcal{T}^{\text{non}}$, respectively. For convenience in computation, we move the items of $\mathbf{K}$ and $\mathbf{V}$ associated with the focal tokens in $\mathcal{T}^{\text{foc}}$ to the front positions. We denote these rearranged items as $\mathbf{K}'$ and $\mathbf{V}'$, respectively. Subsequently, we group and aggregate the $\mathbf{K}'$ and $\mathbf{V}'$ matrices related to the non-focal tokens in $\mathcal{T}^{\text{non}}$ with a group size $m$ based on

their importance in each group, which are formulated as:

$$\mathbf{K}^f = [\mathbf{K}'_1; \dots; \mathbf{K}'_r], \ \mathbf{V}^f = [\mathbf{V}'_1; \dots; \mathbf{V}'_r] \in \mathbb{R}^{r \times d},$$

$$\mathbf{K}^n = \left[ \sum_{j \in G_1} \mathbf{P}_{1,j} \mathbf{K}'_j; \dots; \sum_{j \in G_k} \mathbf{P}_{k,j} \mathbf{K}'_j \right],$$

$$\mathbf{V}^n = \left[ \sum_{j \in G_1} \mathbf{P}_{1,j} \mathbf{V}'_j; \dots; \sum_{j \in G_k} \mathbf{P}_{k,j} \mathbf{V}'_j \right], \quad (14)$$

where $\mathbf{P}_{i,\cdot} = \mathrm{softmax}(\mathbf{Q}_{\max G_i} \mathbf{K}^\top_{[\min G_i : \max G_i]}) \in \mathbb{R}^m$ is the weight vector of $i$-th group, $G_i$ is the index set of $i$-th group, $\min G_i$ and $\max G_i$ are the first and last token index in the $i$-th group, $r = |\mathcal{T}^{\mathrm{foc}}|$ and $k = (L - r)/m$ are the number of focal tokens and groups. Here, we compute the weights $\mathbf{P}$ using only the query of the last token in each group. If non-focal tokens are not divisible by $m$, we will include more tokens in $\mathcal{T}^{\mathrm{foc}}$, ensuring all tokens are processed.

Note that some tokens may not access preceding tokens after grouping aggregation. For instance, in the rightmost of Figure 1, the query $\mathbf{Q}_2$ cannot attend to the keys $\mathbf{K}_1$ and $\mathbf{K}_2$ (*i.e.*, their masks are zeros at the 4-th and 5-th columns) even if the 2-th token should access the 1-th and 2-th tokens. To address this, we introduce complementary keys $\mathbf{K}^c$ and values $\mathbf{V}^c$ to restore the missing information for the query $\mathbf{Q}_i$ (*e.g.*, adding the keys "$\mathbf{K}_1$ and $\mathbf{K}_2$" and the values "$\mathbf{V}_1$ and $\mathbf{V}_2$" to the query $\mathbf{Q}_2$ at the 6-th and 7-th columns):

$$\mathbf{K}^c = [\mathbf{K}_j], \ \mathbf{V}^c = [\mathbf{V}_j], j \in G_z \text{ if } i \in [\min G_z, \max G_z], \quad (15)$$

where $z$ denotes the index of the $i$-th token's nearest neighbor group. We then mask inaccessible keys and values due to the autoregressive nature (more details in Appendix D).

**Group inference.** In the prefilling phase, we can calculate attention in the same way as in the training phase. In the decoding phase, the model generates new tokens, which requires us to dynamically group and aggregate tokens. When the model generates less than $m' = 1.1m$ tokens (with $\sim 10\%$ slots for focal tokens), we use standard attention for calculation; Once $m'$ tokens are generated, we can use the attention of the last query to re-calculate the weight $\mathbf{P}$ and update the group keys $\mathbf{K}^{\mathrm{group}}$ and values $\mathbf{V}^{\mathrm{group}}$ for the subsequent generation. However, we only add a new group key and value to the original ones for reduced memory usage once generating $m'$ tokens.

### 5.2. Fast Focal Token Identification for Grouping

Evaluating the importance of the token is crucial for our group-oriented attention. To this end, we employ an importance score $s$ by computing accumulated attention weights:

$$s_i = \frac{\sum_{j=1}^{L} \mathbf{A}_{j,i}}{\|\mathbf{A}_{\cdot,i}\|_0}, \quad (16)$$

where $\|\mathbf{A}_{\cdot,i}\|_0$ is the number of non-zero elements in the $i$-th column of $\mathbf{A}$, which will be $L - i + 1$ for the decoder-only based transformer. Similar techniques have been used for the inference of LLMs (Zhang et al., 2023; He et al., 2024).

For fast to obtain an importance score $s$, we sample a small portion of recent tokens and some other random tokens to obtain $\widetilde{\mathbf{Q}}$ to approximate the attention weight $\mathbf{A}$:

$$\widetilde{\mathbf{A}} = \mathrm{softmax}\left(\widetilde{\mathbf{Q}} \mathbf{K}^\top / \sqrt{d}\right). \quad (17)$$

For fast implementation, we perform the matrix partitioning technique (Dao et al., 2022) when computing the softmax.

## 6. Experiments

### 6.1. Experimental setup

**Benchmark and evaluation metrics**. We compare our methods on LongBench-E (Bai et al., 2024) to evaluate the long-context understanding capabilities of LLMs, and use EM score (Liu et al., 2024a) to measure the ability to find the key information within a long multi-document context. Additionally, we use inter-token latency (ITL (Chitty-Venkata et al., 2024)) to measure the time delay between generating consecutive tokens. More details are in Appendix C.1.

**Models**. We integrate our DGA attention into LLaMA2-7B (Touvron et al., 2023b) and GPT2-S (Radford et al., 2019) and OPT-125M (Zhang et al., 2022). We train the models on SlimPajama (Cerebras, 2024). All training uses 8 A800 GPUs with a micro-batch size of 1, gradient accumulation of 8, and 1000 steps, consistent across all context lengths.

**Baselines**. We compare our DGA attention with following baselines: StreamingLLM (Xiao et al., 2024), LM-Infinite (Han et al., 2023), and MInference (Jiang et al., 2024). We defer more implementation details for these baseline methods in Appendix C.3.

### 6.2. Empirical Studies of Redundancy in Self-attention

We investigate the redundancy in transformer-based LLMs for long-context modeling by analyzing the sparsity of vanilla attention weights. Figure 2(a) shows the distribution of the vanilla attention weights for a sequence randomly sampled from SlimPajama on LLaMA-7B, which shows that only a few tokens contribute significantly to the predictions.

To further quantify the sparsity across more examples on long contexts, we examine $P_{sparse}(L, \rho)$ of the attention weights across 100 contexts of length 1k over SlimPajama on LLaMA-7B at layers 24 and 30 under different sparse rates $\rho$. Note that lower $\rho$ means sparser attention weights. Figure 2(b) shows that, as $\rho$ decreases, *i.e.*, $\rho : 0.05 \rightarrow 0.02 \rightarrow 0.01$, the probability $P_{sparse}(L, \rho)$ increases with context length $L$, indicating that attention

*Table 1.* Comparisons of different methods on LongBench-E (Bai et al., 2024), where the 95% text length quantile is 31K. "ITL" denotes inter-token latency (Chitty-Venkata et al., 2024), which measures the time delay between generating consecutive tokens.

| Methods | Single-Doc. QA | Multi-Doc. QA | Summary | FS. Learning | Synthetic | Code | Avg. ↑ | ITL / ms ↓ |
|---|---|---|---|---|---|---|---|---|
| *LLaMA2-7B (Vanilla Self-Attention)* | 6.43 | 2.37 | 13.65 | 56.65 | 3.04 | 48.0 | **21.69** | 69.70 |
| MInference (Jiang et al., 2024) | 5.86 | 2.65 | 14.33 | 55.99 | 2.63 | 48.41 | 21.64 | 94.34 |
| StreamingLLM (Xiao et al., 2024) | 4.99 | 4.13 | 11.51 | 45.43 | 2.16 | 30.38 | 16.43 | 78.28 |
| LM-Infinite (Han et al., 2023) | 3.54 | 2.61 | 3.31 | 48.97 | 1.33 | 35.26 | 15.84 | 102.22 |
| DGA-LLM (Ours) | 3.61 | 3.58 | 6.81 | 57.90 | 1.47 | 53.45 | 21.14 | **28.80** |

*Table 2.* Comparisons of different methods on EM score (Liu et al., 2024a) evaluated at lengths from 4K to 32K.

| Methods | 4K | 8K | 16K | 32K | Avg. |
|---|---|---|---|---|---|
| *LLaMA2-7B (Vanilla Self-Attention)* | 37.2 | 36.4 | 33.8 | 26.8 | **33.6** |
| StreamingLLM | 30.2 | 25.8 | 22.2 | 20.8 | 24.8 |
| LM-Infinite | 29.4 | 28.6 | 23.8 | 22.4 | 26.1 |
| MInference | 29.2 | 24.8 | 23.6 | 17.0 | 23.7 |
| DGA-LLM (Ours) | 35.0 | 27.4 | 25.6 | 22.6 | 27.7 |

*Table 3.* Comparisons of different methods on computational efficiency in terms of inter-token latency (ITL (Chitty-Venkata et al., 2024)) evaluated at lengths from 4K to 16K.

| Methods | 4K | 8K | 16K |
|---|---|---|---|
| *LLaMA2-7B (Vanilla Self-Attention)* | 36.87 | 42.32 | 69.70 |
| MInference | 93.13 | 93.93 | 94.34 |
| StreamingLLM | 40.00 | 46.67 | 78.28 |
| LM-Infinite | 49.70 | 64.34 | 102.22 |
| DGA-LLM (Ours) | **26.26** | **26.87** | **28.79** |

weights become increasingly sparse with longer contexts, and only a few tokens contribute significantly to predictions.

Note that $P_{sparse}(L, \rho)$ is dataset-independent and adapts to $L$. Figure 2(c) shows $P_{sparse}$ has *nearly identical trends* across SlimPajama and WikiText2 over llama2-7B or Qwen2.5-7B model, confirming $\rho$ generalizes across datasets. As $\rho = 0.02$, $P_{sparse}$ increases sharply with $L$, where $P_{sparse} \rightarrow 1$ at $L = 400$ over llama2-7B and $L = 300$ over Qwen2.5-7B, respectively, proving sparsity strengthens with context length universally and model-level characterized. These results align with our analysis in Theorem 1, highlighting the inherent redundancy in long-context attention computations. More results are in Appendix H.1.

### 6.3. Comparisons on Long-Context Modeling

**Results on Longbench-E.** We conduct experiments on LongBench-E (Bai et al., 2024), which includes tasks with text lengths up to 31K. Table 1 shows that our DGA-LLM achieves competitive performance across tasks, with an average score of 21.14 for LLaMA2-7B trained on 8K context-length texts. While our method does not significantly outperform others in average scores, it demonstrates notable advantages in Inter-token Latency (ITL) at 16K context length. Specifically, DGA-LLM achieves an ITL of 28.80 ms, $2.42\times$ faster than vanilla LLaMA2-7B (69.70 ms) and $3.55\times$ faster than LM-Infinite (102.22 ms). This highlights

the efficiency of our group coding strategy in reducing computational redundancy while maintaining performance, making it particularly suitable for time-sensitive applications. More results on ITL are in Section 6.4.

**Results on EM score.** We further validate the effectiveness of our DGA-LLM on long-document EM scores at context lengths of 4K, 8K, 16K and 32K for LLaMA2-7B trained with a context length of 8K. As shown in Table 2, our DGA-LLM achieves the highest performance at 4K, 16K, and 32K context lengths, and the second highest at 8K compared to other baselines, with average scores of 27.7. In contrast, methods like StreamingLLM and LM-Infinite exhibit significant performance degradation as context length increases, particularly at 16K, where their scores drop to 22.2 and 23.8, respectively. These results highlight that our DGA-LLM not only reduces computational redundancy but also maintains competitive performance, making it highly suitable for diverse long-context scenarios.

**Results on optimization efficiency.** To validate the optimization efficiency of the group coding strategy in DGA, we pre-train GPT2-S and OPT-125M with a 1K and 2K context length on SlimPajama, respectively, as fine-tuning LLMs from scratch is infeasible. As shown in Figures 3(a) and 4, the validation loss for our DGA converges faster than vanilla self-attention, indicating that our group coding strategy accelerates convergence. These results align with the conclusion in Theorem 3, demonstrating the efficiency of our approach in optimizing long-context modeling tasks.

**Results on robustness to random noise.** To evaluate the robustness of the group coding strategy in DGA to random noise, we conduct experiments on question answering task for LLaMA2-7B trained with context-length 8K and compare the average KL-divergence of the output probability before and after adding the noises on random sampled 100 sequences on SlimPajama. From Figure 3(b), vanilla self-attention exhibits a significant deviation under noisy conditions, while our DGA maintains a more robust KL-divergence. This coincides with Theorem 2, demonstrating that group coding enhances robustness to random noise.

### 6.4. Comparisons on Computational Efficiency

To evaluate the inference efficiency of our DGA, we measure the Inter-token Latency (ITL) of the LLaMA2-7B model

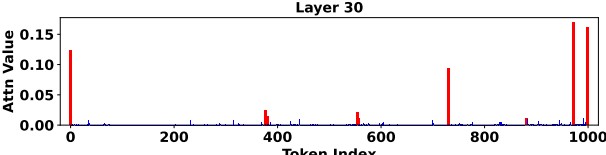

(a) Distribution of attention weights for a random context.

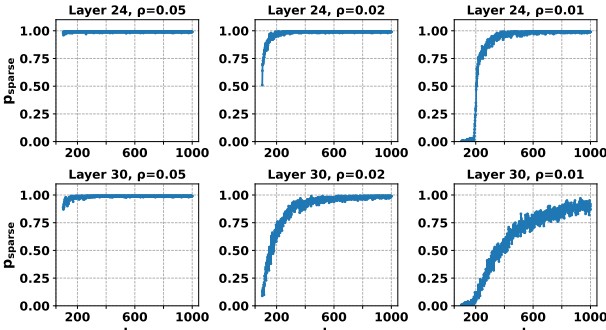

(b) Distribution of $\rho$-sparse across different context lengths.

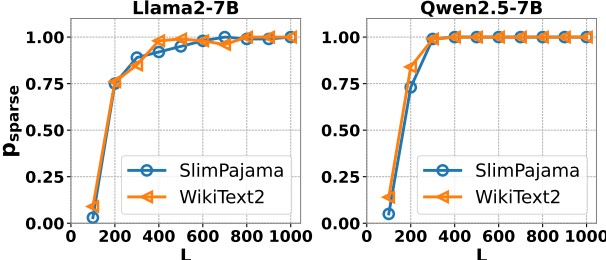

(c) Distribution of $\rho$-sparse across different datasets and models.

*Figure 2.* Sparsity on the attention weights on for long-context modeling. (a) shows the attention weight distribution for a random example on SlimPajama. (b) demonstrates the distribution of $\rho$-sparse across context lengths, *i.e.*, $P_{sparse}(L, \rho)$, over 100 random examples on SlimPajama. (c) exhibits the distribution of $\rho$-sparse over 100 random examples on WikiText2 and SlimPajama, across llama2-7B (left) and Qwen2.5-7B models (right) as $\rho = 0.02$.

during the decoding phase across 4K, 8K, and 16K contexts on LongBench-E using a single A800 GPU. Table 3 shows that DGA significantly reduces inference latency compared to other methods. For instance, at 16K context length, DGA-LLM achieves an ITL of 28.79 ms, $3.28\times$ faster than MInference (94.34 ms) and $2.72\times$ faster than StreamingLLM (78.28 ms). Notably, other methods are slower than vanilla self-attention, as they lack optimizations for the generation process. As the context length increases from 4K to 16K, other baseline methods show a sharp ITL rise, indicating a slower generation. In contrast, our DGA-LLM maintains stable and low ITL, demonstrating its efficiency for longer texts (see more detailed complexity analysis in Appendix E). Notably, LongLoRA (Chen et al., 2024) uses $S^2$-Attention only during training, reverting to vanilla self-attention in inference, resulting in the same speed. These results highlight the efficiency of DGA-LLM in handling long-context tasks.

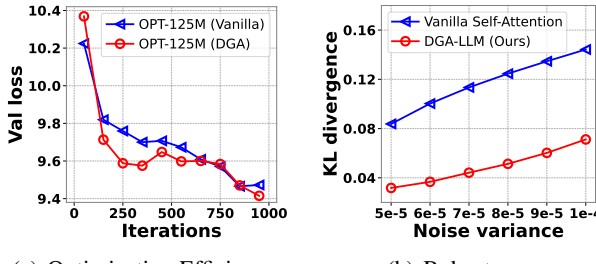

(a) Optimization Efficiency    (b) Robustness

*Figure 3.* Comparisons on optimization efficiency and robustness to random noise between Vanilla Self-Attention and DGA-LLM (ours). Subfigure (a) shows validation losses of our DGA-LLM and vanilla ones on OPT-125M, where the models are trained with a 2K context length on SlimPajama. Subfigure (b) demonstrates average KL-divergence between the output probability distributions before and after adding Gaussian noise for Vanilla Self-Attention and our DGA-LLM under different levels of noise variance ($\sigma^2$).

## 7. Conclusion

In this paper, we address computational redundancy in long-context modeling by first reformulating the sequence modeling as a supervised learning task, enabling a clear understanding of token relevance. Based on this reformulation, we theoretically analyze attention sparsity, showing that only a few tokens significantly contribute to predictions, and propose a group coding strategy. Last, we propose Dynamic Group Attention (DGA), which uses this strategy to adaptively reduce redundancy while retaining critical token interactions. We validate the effectiveness of DGA on long-context tasks, showing significant reductions in computational costs while maintaining competitive performance.

## Acknowledgments

This work was partially supported by the Joint Funds of the National Natural Science Foundation of China (Grant No.U24A20327), Key-Area Research and Development Program Guangdong Province 2018B010107001, Postdoctoral Fellowship Program of CPSF (GZC20251043), and TCL Science and Technology Innovation Fund, China.

## Impact Statement

Our work addresses the computational inefficiency of long-context modeling in LLMs by introducing a dynamic token grouping strategy. By reducing redundant attention computations while preserving critical interactions, our method lowers training and inference costs, making long-context applications (*e.g.*, legal document analysis, multi-hop QA) more accessible for resource-constrained settings. Our approach contributes to sustainable AI development by balancing performance and efficiency, paving the way for scalable and practical deployment of LLMs in real-world scenarios.

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

# APPENDIX

## Contents

# A. Theoretical analysis

### A.1. Proof of Theorem 1

**Theorem 1** *(Sparsity on attention weights) Consider $\rho \in (1/L, 1]$ as a sparse rate, we say that the weight $\alpha$ is $\rho$-sparse when there exists at least one probability greater than $1/(L\rho)$. Let $\xi = \mathbf{K}\mathbf{Q}_i \in \mathbb{R}^L$, then the probability of $\alpha$ being $\rho$-sparse $P_{sparse}(L, \rho)$ is given by*

$$P_{sparse}(L, \rho) \geq \max_{x>0} 1 - [P_{head}P_{tail}]^L, \tag{18}$$

*where $P_{head} = P\{\exp(\xi_j) \leq x\}$ with $j$ being some index of $\alpha$ and $P_{tail} = P\{(L\rho - 1)x \leq \sum_{k \neq j}^L \exp(\xi_k)\}$.*

*Proof.* According to the definition of $\rho$-sparse, the probability of having at least one probability greater than $1/(L\rho)$ is given

$$\begin{aligned} P_{sparse}(L, \rho) &= 1 - P\{\alpha_j \leq 1/(L\rho), \forall j \in [L]\} \\ &\geq 1 - P^L\{\alpha_j \leq 1/(L\rho)\} \end{aligned} \tag{19}$$

Here, the inequation is due to the dependence of each $\alpha_j$. We next focus on the upper bound of $P\{\alpha_j \leq 1/(L\rho)\}$.

$$\begin{aligned} P\{\alpha_j \leq 1/(L\rho)\} &= P\{\langle \exp(\xi), \mathbf{1}_n \rangle^{-1} \exp(\xi_j) \leq 1/(L\rho)\} \\ &= P\{\exp(\xi_j) \leq 1/(L\rho)\langle \exp(\xi), \mathbf{1}_n \rangle\} \\ &= P\{\exp(\xi_j) \leq x \leq 1/(L\rho)\langle \exp(\xi), \mathbf{1}_n \rangle\} \\ &= P\{\exp(\xi_j) \leq x, x \leq 1/(L\rho)\langle \exp(\xi), \mathbf{1}_n \rangle\} \\ &= P\{\exp(\xi_j) \leq x\}P\{x \leq 1/(L\rho)\langle \exp(\xi), \mathbf{1}_n \rangle | \exp(\xi_j) \leq x\} \\ &= P\{\exp(\xi_j) \leq x\}P\{L\rho x \leq \langle \exp(\xi), \mathbf{1}_n \rangle | \exp(\xi_j) \leq x\} \\ &\leq P\{\exp(\xi_j) \leq x\}P\{(L\rho - 1)x \leq \sum_{k \neq j}^n \exp(\xi_k)\} \end{aligned} \tag{20}$$

Combining the inequations (19) and (20), and denoting $P_{head} = P\{\exp(\xi_j) \leq x\}$ and $P_{tail} = P\{(L\rho - 1)x \leq \sum_{k \neq j}^L \exp(\xi_k)\}$, we finish the proof. $\qquad\square$

Note that the distribution of the attention weights $\alpha$ is often complex, and more importantly, these weights are not independent and may be highly correlated, which makes the analysis of sparsity extremely challenging. In this theorem, we provide a general bound for sparsity without imposing specific assumptions on the weights. In the inequation (18), a larger $x$ increases $P_{head}$ but decreases $P_{tail}$, ensuring that their product remains bounded by a value less than 1. When the sequence length $L$ is large, this value raised to the power of $L$ will be much less than 1, resulting in a large $P_{sparse}(L, \rho)$. We hope this theorem offers researchers deeper insights and serves as a source of inspiration for further exploration into sparsity.

### A.2. Proof of Theorem 2

**Theorem 2** *Assume the weights before normalization are $\tilde{\alpha}$ obtained by the product of the query and key matrix, and $\alpha_j = \frac{\exp(\tilde{\alpha}_j)}{\sum_{l=1}^L \exp(\tilde{\alpha}_l)}$. Consider a Gaussian noise $\Delta\tilde{\alpha} \sim \mathcal{N}(\mathbf{0}, \sigma^2 \mathbf{I}_L)$ being added into $\tilde{\alpha}$ and each group size being $|G_g| = m$, the variance of the weight changes obtained via group optimization in Eqn. (2) can be reduced by $1/m^2$.*

*Proof.* Recall that the optimization in Problem (2) when $|G_g| = m$ is:

$$\min_{\bar{\alpha}} \left\| \sum_{g=1}^k \sum_{j \in G_g} \frac{\bar{\alpha}_g}{m} \mathbf{V}_j - \mathbf{y} \right\|_2^2, \text{ s.t., } \sum_{g=1}^k m\alpha_g = 1, \ \alpha_g > 0. \tag{21}$$

Note that adding the noise $\Delta\tilde{\alpha}_g \sim \mathcal{N}(0, \sigma^2)$ on $\bar{\alpha}_g$ is equivalent to adding the noise $\frac{\tilde{\alpha}_g}{m}$ on each weight in $g$-th group. Consequently, we focus only on the effect of the input noise $\frac{\tilde{\alpha}_g}{m}$ on the normalized output. Since the factor $\frac{1}{m}$ can be shifted on the variance $\sigma$ of the noise, we next direct analyze the effect of the input noise $\tilde{\alpha}_j$ on the normalized $\alpha_j$ for simplicity.

Denote the normalized noised weight as

$$\alpha_j^\delta = \frac{\exp(\tilde{\alpha}_j^\delta)}{\sum_{i=1}^L \exp(\tilde{\alpha}_i^\delta)}, \tag{22}$$

where $\alpha_j^\delta = \tilde{\alpha}_j + \Delta\tilde{\alpha}_j$ and $\Delta\tilde{\alpha}_j \sim \mathcal{N}(0, \sigma^2)$. Then, according to Lemma 1 and Lemma 2, we have

$$\mathrm{Var}(\Delta\alpha_j) = \mathrm{Var}(\alpha_j^\delta - \alpha_j) \tag{23}$$

$$= \mathrm{Var}\left[\alpha_j(\Delta\hat{\alpha}_j - \sum_{j=1}^L \alpha_j \Delta\tilde{\alpha}_j + o(\Delta\tilde{\alpha}_j))\right]$$

$$= \alpha_j^2 \, \mathrm{Var}\left[\Delta\hat{\alpha}_j - \sum_{j=1}^L \alpha_j \Delta\tilde{\alpha}_j + o(\Delta\tilde{\alpha}_j)\right]$$

$$= \mathcal{O}(\sigma^2) \tag{24}$$

where $o(\Delta\tilde{\alpha}_j)$ represents the higher-order infinitesimal of $\Delta\tilde{\alpha}_j$ and $\mathcal{O}(\sigma^2)$ denotes the same-order infinitesimal of $\sigma^2$ (Browder, 2012). Thus, when adding the noise $\frac{\tilde{\alpha}_g}{m}$ to each weight in the $g$-th group, the variance of the weight changes is $\mathcal{O}(\sigma^2/m^2)$. Therefore, by group optimization in Eqn. (2), the variance of the weight changes can be reduced by $1/m^2$. $\square$

**Lemma 1.** *Given* $\alpha_j = \frac{\exp(\tilde{\alpha}_j)}{\sum_{l=1}^L \exp(\tilde{\alpha}_l)}$ *and* $\alpha_j^\delta = \frac{\exp(\tilde{\alpha}_j + \Delta\tilde{\alpha}_j)}{\sum_{i=1}^L \exp(\tilde{\alpha}_i + \Delta\tilde{\alpha}_i)}$, *where* $\Delta\tilde{\alpha}_j \sim \mathcal{N}(0, \sigma^2)$, *let* $\Delta\alpha_j = \alpha_j^\delta - \alpha_j$, *we have*

$$\Delta\alpha_j = \alpha_j \left[\Delta\tilde{\alpha}_j - \sum_{i=1}^L \alpha_i \Delta\tilde{\alpha}_i + o(\Delta\tilde{\alpha}_j)\right], \tag{25}$$

*where* $o(\Delta\tilde{\alpha}_j)$ *represents the higher-order infinitesimal of* $\Delta\tilde{\alpha}_j$ *(Browder, 2012).*

*Proof.* According to the definition of $\Delta\tilde{\alpha}_j$, we have

$$\alpha_j^\delta = \frac{\exp(\tilde{\alpha}_j + \Delta\tilde{\alpha}_j)}{\sum_{i=1}^L \exp(\tilde{\alpha}_i + \Delta\tilde{\alpha}_i)}$$

$$= \frac{\exp(\tilde{\alpha}_j)[(1 + \Delta\tilde{\alpha}_j + o(\Delta\tilde{\alpha}_j)]}{\sum_{i=1}^L \exp(\tilde{\alpha}_i)[1 + \Delta\tilde{\alpha}_i + o(\Delta\tilde{\alpha}_i)]}$$

$$= \frac{\exp(\tilde{\alpha}_j)[(1 + \Delta\tilde{\alpha}_j + o(\Delta\tilde{\alpha}_j)]}{\sum_{n=1}^L \exp(\tilde{\alpha}_n) + \sum_{i=1}^L \exp(\tilde{\alpha}_i)[\Delta\tilde{\alpha}_i + o(\Delta\tilde{\alpha}_i)]}$$

$$= \frac{\exp(\tilde{\alpha}_j)[(1 + \Delta\tilde{\alpha}_j + o(\Delta\tilde{\alpha}_j)]}{\sum_{n=1}^L \exp(\tilde{\alpha}_n)\left\{1 + \sum_{i=1}^L \frac{\exp(\tilde{\alpha}_i)}{\sum_{m=1}^L \exp(\tilde{\alpha}_m)}[\Delta\tilde{\alpha}_i + o(\Delta\tilde{\alpha}_i)]\right\}}$$

$$= \frac{\exp(\tilde{\alpha}_j)[(1 + \Delta\tilde{\alpha}_j + o(\Delta\tilde{\alpha}_j)]}{\sum_{n=1}^L \exp(\tilde{\alpha}_n)\left\{1 + \sum_{i=1}^L \alpha_i[\Delta\tilde{\alpha}_i + o(\Delta\tilde{\alpha}_i)]\right\}}$$

$$= \frac{\alpha_j[(1 + \Delta\tilde{\alpha}_j + o(\Delta\tilde{\alpha}_j)]}{1 + \sum_{i=1}^L \alpha_i[\Delta\tilde{\alpha}_i + o(\Delta\tilde{\alpha}_i)]}$$

$$= \alpha_j\left[(1 + \Delta\tilde{\alpha}_j + o(\Delta\tilde{\alpha}_j)]\left[1 - \sum_{i=1}^L \alpha_i[\Delta\tilde{\alpha}_i + o(\Delta\tilde{\alpha}_i)] + o\left(\sum_{i=1}^L \alpha_i[\Delta\tilde{\alpha}_i + o(\Delta\tilde{\alpha}_i)]\right)\right]\right]$$

$$= \alpha_j\left[1 + \Delta\tilde{\alpha}_i - \sum_{i=1}^L \alpha_i \Delta\tilde{\alpha}_i + o(\Delta\tilde{\alpha}_i)\right] \tag{26}$$

where the second line follows Taylor's expansion of $\exp(\Delta\tilde{\alpha}_j) = 1 + \Delta\tilde{\alpha}_j + o(\Delta\tilde{\alpha}_j)$; the penultimate line follows Taylor's expansion of $\frac{1}{1 + \sum_{i=1}^L \alpha_i[\Delta\tilde{\alpha}_i + o(\Delta\tilde{\alpha}_i)]} = 1 - \sum_{i=1}^L \alpha_i[\Delta\tilde{\alpha}_i + o(\Delta\tilde{\alpha}_i)] + o(\sum_{i=1}^L \alpha_i[\Delta\tilde{\alpha}_i + o(\Delta\tilde{\alpha}_i)])$.

Therefore, we have

$$\Delta\alpha_j = \alpha_j^\delta - \alpha_j = \alpha_j \left[ \Delta\tilde{\alpha}_j - \sum_{i=1}^{L} \alpha_i \Delta\tilde{\alpha}_i + o(\Delta\tilde{\alpha}_j) \right]. \tag{27}$$

$\square$

**Lemma 2.** *Given $\Delta\tilde{\alpha} \sim \mathcal{N}(\mathbf{0}, \sigma^2 \mathbf{I}_L)$, we have*

$$\mathrm{Var}(\Delta\tilde{\alpha}_j - \sum_{i=1}^{L} \alpha_i \Delta\tilde{\alpha}_i) = \mathcal{O}(\sigma^2), \tag{28}$$

*where $\mathcal{O}(\sigma^2)$ denotes the same-order infinitesimal of $\sigma^2$ ([Browder, 2012](#)).*

*Proof.* Expanding the variance of $\Delta\alpha_j - \sum_{i=1}^{L} \alpha_i \Delta\tilde{\alpha}_i$ as

$$\mathrm{Var}(\Delta\alpha_j - \sum_{i=1}^{L} \alpha_i \Delta\tilde{\alpha}_i) = \mathrm{Var}(\Delta\tilde{\alpha}_j) + \mathrm{Var}\left( \sum_{i=1}^{L} \alpha_i \Delta\tilde{\alpha}_i \right) - 2 \cdot \mathrm{Cov}\left( \Delta\tilde{\alpha}_j, \sum_{i=1}^{L} \alpha_i \Delta\tilde{\alpha}_i \right)$$

$$= \sigma^2 + \sigma^2 \sum_{i=1}^{L} \alpha_i^2 - 2 \sum_{i=1}^{L} \alpha_i \cdot \mathrm{Cov}(\Delta\tilde{\alpha}_j, \Delta\tilde{\alpha}_i)$$

$$= \sigma^2 + \sigma^2 \sum_{i=1}^{L} \alpha_i^2 - 2\alpha_j \sigma^2$$

$$= (1 + \sum_{i=1}^{L} \alpha_i^2 - 2\alpha_j)\sigma^2$$

$$= \mathcal{O}(\sigma^2) \tag{29}$$

where the third line holds since $\mathrm{Cov}(\Delta\tilde{\alpha}_i, \Delta\tilde{\alpha}_j) = 0$ when $i \neq j$ and $\mathrm{Cov}(\Delta\tilde{\alpha}_i, \Delta\tilde{\alpha}_j) = \sigma^2$ when $i \neq j$. $\square$

### A.3. Proof of Theorem 3

**Theorem 3** *Denote $H$ and $\bar{H}$ as the Hessian matrix of the optimizations in ([1](#)) and ([2](#)), respectively. Let $\kappa(H) = \lambda_{\max}(H)/\lambda_{\min}(H)$ be the condition number ([Edelman, 1988](#)) of $H$, where $\lambda_{\max}$ and $\lambda_{\min}$ are the maximum and minimum eigenvalues of $H$, respectively. Consider each group size $|G_g| = m$ and $\lambda_{\min}(H) > 0$, we have*

$$\kappa(\bar{H}) \leq \kappa(H). \tag{30}$$

*Proof.* Recall that the objective in Problem ([2](#)) when $|G_g| = m$ is:

$$\min_{\bar{\alpha}} \left\| \sum_{g=1}^{k} \sum_{j \in G_g} \frac{\bar{\alpha}_g}{m} \mathbf{V}_j - \mathbf{y} \right\|_2^2, \text{ s.t., } \sum_{g=1}^{k} m\alpha_g = 1, \ \alpha_g > 0. \tag{31}$$

Let $\bar{\mathbf{V}}_g = \frac{1}{m} \sum_{j \in G_g} \mathbf{V}_j$ and $\bar{\mathbf{V}} = [\bar{\mathbf{V}}_1; \bar{\mathbf{V}}_2; \dots; \bar{\mathbf{V}}_k] \in \mathbb{R}^{k \times d}$, then the Hessian matrix of the grouped objective is

$$\bar{H} = 2\bar{\mathbf{V}}^\top \bar{\mathbf{V}}. \tag{32}$$

We introduce a grouping matrix $M \in \mathbb{R}^{k \times L}$, where $M_{g,i}$ is defined as

$$M_{g,i} = \begin{cases} \frac{1}{m}, & i \in g, \\ 0, & i \notin g. \end{cases} \tag{33}$$

Then, we have

$$\bar{\mathbf{V}} = M\mathbf{V}. \tag{34}$$

The Hessian matrix in (32) can be reformulated as

$$\bar{H} = 2\bar{\mathbf{V}}^\top \bar{\mathbf{V}} = 2\mathbf{V}^\top M^\top M\mathbf{V}. \tag{35}$$

According to Lemma 3 and 4, we have

$$\frac{\lambda_{\max}(\bar{H})}{\lambda_{\min}(\bar{H})} \leq \frac{\lambda_{\max}(H)}{\lambda_{\min}(H)}. \tag{36}$$

Thus, based on the definition of the condition number (Edelman, 1988), we have

$$\kappa(\bar{H}) \leq \kappa(H). \tag{37}$$

$\square$

**Lemma 3.** *Given $\bar{H} = 2\mathbf{V}^\top M^\top M\mathbf{V}$ and $H = 2\mathbf{V}^\top \mathbf{V}$, where $M$ is defined in Eqn. (33), we have*

$$\lambda_{\max}(\bar{H}) \leq \frac{\lambda_{\max}(H)}{m}. \tag{38}$$

*Proof.* Note that the matrix $M^\top M$ is a block-diagonal matrix:

$$M^\top M = \begin{bmatrix} \frac{1}{m^2} J_m & 0 & \cdots & 0 \\ 0 & \frac{1}{m^2} J_m & \cdots & 0 \\ \vdots & \vdots & \ddots & \vdots \\ 0 & 0 & \cdots & \frac{1}{m^2} J_m \end{bmatrix} \tag{39}$$

where $J_m \in \mathbb{R}^{m \times m}$ is an all-one matrix.

Note that the rank of $J_m$ is $\text{rank}(J_m) = 1$ since each row of the matrix is 1. Due to $J_m \mathbf{v} = m\mathbf{v}$, where $\mathbf{v} = \frac{1}{\sqrt{m}}[1, 1, \cdots, 1]^\top$, one of the (non-zero) eigenvalues of $\frac{1}{m^2} J_m$ is

$$\lambda_{\max}(\frac{1}{m^2} J_m) = \frac{m}{m^2} = \frac{1}{m}. \tag{40}$$

Thus, we have

$$\lambda_{\max}(M^\top M) = \frac{1}{m}. \tag{41}$$

Based on the property of the spectral norm of the matrix (Mathias, 1990), we have

$$S(\mathbf{V}^\top M^\top M\mathbf{V}) \leq S(\mathbf{V}^\top \mathbf{V}) \cdot S(M^\top M), \tag{42}$$

where $S(A)$ is the spectral norm of the matrix $A$. Since the spectral norm of the symmetric matrix $A$ is its maximum eigenvalue, we have

$$\lambda_{\max}(\mathbf{V}^\top M^\top M\mathbf{V}) \leq \lambda_{\max}(\mathbf{V}^\top \mathbf{V}) \cdot \lambda_{\max}(M^\top M). \tag{43}$$

Combing Eqn. (41) and the definitions of $\bar{H}$ and $H$, we get

$$\lambda_{\max}(\bar{H}) \leq \frac{\lambda_{\max}(H)}{m}. \tag{44}$$

$\square$

**Lemma 4.** *Given $\bar{H} = 2\mathbf{V}^\top M^\top M\mathbf{V}$ and $H = 2\mathbf{V}^\top \mathbf{V}$, where $M$ is defined in Eqn. (33), we have*

$$\lambda_{\min}(\bar{H}) \geq \frac{\lambda_{\min}(H)}{m}. \tag{45}$$

*Proof.* According to the Rayleigh theorem (Horn & Johnson, 2012), for $M^\top M$, for any nonzero vector $\mathbf{z} \in \mathbb{R}^L$, we have

$$\frac{\mathbf{z}^\top M^\top M \mathbf{z}}{\mathbf{z}^\top \mathbf{z}} \geq \lambda_{\min}(M^\top M). \tag{46}$$

Let $\mathbf{z} = \mathbf{V}\mathbf{x}$, where $x \in \mathbb{R}^d$ and $\|\mathbf{x}\|_2^2 = 1$, we have

$$\mathbf{x}^\top \mathbf{V}^\top M^\top M \mathbf{V}\mathbf{x} \geq \lambda_{\min}(M^\top M) \cdot \mathbf{x}^\top \mathbf{V}^\top \mathbf{V}\mathbf{x}. \tag{47}$$

Taking the minimum of all the vectors $x$, we get

$$\min_{\|\mathbf{x}\|_2^2=1} \mathbf{x}^\top \mathbf{V}^\top M^\top M \mathbf{V}\mathbf{x} \geq \lambda_{\min}(M^\top M) \cdot \min_{\|\mathbf{x}\|_2^2=1} \mathbf{x}^\top \mathbf{V}^\top \mathbf{V}\mathbf{x}. \tag{48}$$

According to the definition of the minimum eigenvalues, we have

$$\lambda_{\min}(\mathbf{V}^\top M^\top M \mathbf{V}) \geq \lambda_{\min}(M^\top M) \cdot \lambda_{\min}(\mathbf{V}^\top \mathbf{V}). \tag{49}$$

Here, we focus on the non-zero eigenvalues of $\mathbf{V}^\top M^\top M \mathbf{V}$ since the zero eigenvalues may not actually mean much. Based on the analyses in the eigenvalues of $M^\top M$ in Lemma 3, we have $\lambda_{\min}(M^\top M) = \frac{1}{m}$. Thus, we obtain

$$\lambda_{\min}(\bar{H}) \geq \frac{\lambda_{\min}(H)}{m}. \tag{50}$$

$\square$

## B. More Discussions on Sequence Modeling and Supervised Learning

We demonstrate that traditional pre-training is fundamentally a form of supervised learning. For a token sequence $\mathbf{x}_1, \mathbf{x}_2, \ldots, \mathbf{x}_{t-1}, \mathbf{x}_t$, traditional pre-training, based on the principle of maximum likelihood, optimizes the model $\theta$ as

$$\max_{\theta} P_\theta(\mathbf{x}_1, \mathbf{x}_2, \ldots, \mathbf{x}_{t-1}, \mathbf{x}_t) = \prod_{k=1}^{t} p(\mathbf{x}_k | \mathbf{x}_1, \mathbf{x}_2, \ldots, \mathbf{x}_{k-1}), \mathbf{x}_k \in \mathcal{X}. \tag{51}$$

Given a large collection of sequences, this objective aligns with Eqn. 2, with the conditions represented as $C(\mathbf{y})$. In practical applications, this can be reformulated as minimizing the cross-entropy loss:

$$\min_{\theta} - \sum_{k} \log p(\mathbf{x}_k | \mathbf{x}_1, \mathbf{x}_2, \ldots, \mathbf{x}_{k-1}). \tag{52}$$

Thus, the traditional pre-training paradigm for LLMs is essentially equivalent to supervised learning.

# C. More Details for Experiment Settings

## C.1. More Details on Benchmark and Evaluation Metrics

**SlimPajama** (Cerebras, 2024) dataset is an open-source reproduction of the data mixture used to pretrain the LLaMA models. It includes 82% web data, 4.5% code from GitHub, 4.5% Wikipedia, 4.5% books, 2.5% Arxiv, and 2.0% StackExchange. The dataset is designed to extend the context lengths of large language models (LLMs) to 128K tokens through data engineering techniques such as per-source length upsampling.

**LongBench** (Bai et al., 2024) is a pioneering benchmark designed to evaluate the long-context understanding capabilities of large language models (LLMs) in a bilingual (Chinese and English) and multitask setting. It includes 21 tasks across 6 major categories, such as single-document QA, multi-document QA, summarization, few-shot learning, synthetic tasks, and code completion, covering key long-text application areas. The benchmark comprises 14 English tasks, 5 Chinese tasks, and 2 code tasks, with most tasks averaging between 5k to 15k in length and totaling 4,750 test data points. LongBench contains a test set with more evenly distributed lengths, named LongBench-E, which includes comparable data quantities across 0-4K, 4K-8K, and 8K+ length intervals, enabling thorough analysis of model performance across different input lengths.

**EM score** (Liu et al., 2024a) measures the ability to find the key information within a long multi-document context. It measures whether the model's predicted output exactly matches one or more ground-truth answers, ensuring precision in information retrieval. Therefore, the EM score is calculated by checking if any correct answers appear in the model's predictions, providing a strict assessment of the model's ability to extract precise information from complex contexts. This metric is particularly stringent, as it requires the model to identify the correct answer without partial or approximate matches.

**Inter-token latency** (ITL (Chitty-Venkata et al., 2024)) is a metric for evaluating the average time between generating consecutive tokens in a response. It refers to the time delay between the generation of consecutive tokens (e.g., words or subwords) in a sequence produced by a language model. This metric is influenced by factors like model architecture, computational resources, and optimization techniques. Reducing inter-token latency is essential for enhancing the usability and performance of language models in time-sensitive scenarios.

**Perplexity (PPL)** serves as a metric to evaluate a model's predictive accuracy within a given context. It is derived by exponentiating the average negative log-likelihood of a sequence, providing a statistical assessment of language modeling efficacy. **Proof-pile** (Azerbayev et al., 2022) is a high-quality dataset encompassing mathematical text and code, totaling 13GB and consisting of 8.3 billion tokens (tokenized using the gpt-neox tokenizer). This dataset aggregates a variety of sources, including both informal and formal mathematical content, with raw data collected from the web. The PPL is reported based on the test dataset. We use the test dataset of Proof-pile to verify the modeling ability of models.

## C.2. Implementation Details on Our Method

For continuous pretraining, we utilize the SlimPajama dataset, an open-source replication of the LLaMA pretraining data mixture. In our approach, we replace the standard self-attention mechanism in LLaMA-2 with our focal attention. Notably, the group size $m = 16$ and importance rate $\gamma = 0.1$, the number of the focal tokens is $\max\{\gamma L, 1k\}$, remain consistent across different experiments, ensuring architectural uniformity. All models are trained on $8 \times$ A800 GPUs, employing a micro-batch size of 1 with gradient accumulation over 8 steps, totaling 1000 training steps. To facilitate effective scaling to longer contexts, we modified the RoPE (Rotary Position Embedding) base frequency from 10,000 to 500,000, following Cerebras (2024) and Xiong et al. (2024).

## C.3. Implementation Details on Baselines

We implement StreamingLLM (Xiao et al., 2024) following the codebase[1], LM-Infinite (Han et al., 2023) following the codebase[2], MInference (Jiang et al., 2024) following the codebase[3].

For StreamingLLM, we configure the attention sink parameter to 4 and the attention context window size to 2000 tokens. For LM-Infinite, the local branch size is set to 4096 tokens, while the global branch size is configured to 10. For MInference, we utilize their official implementations with default parameter settings.

---

[1] https://github.com/mit-han-lab/streaming-llm
[2] https://github.com/Glaciohound/LM-Infinite
[3] https://github.com/microsoft/MInference

## C.4. Implementation Details on Sparsity in Figure 2

In Section 6.2, we conduct experiments on the sparsity of the vanilla attention weights on the samples from the train split of the SlimPajama (Cerebras, 2024) dataset, truncating each sample to retain only the first 1,000 tokens. Each sample undergoes a single forward pass through the LLaMA2-7B model (Touvron et al., 2023b), during which we capture the attention maps from the 9-th attention head of the 24-th and 30-th layers. In Figure 2(a), we randomly select a example and display the distribution of attention weights corresponding to the last token in the 30-th layer. Additional results are provided in Figure 6. In Figure 2(b), we randomly select 100 examples and present the statistical analysis of attention distributions across rows $L \in [100, 1000]$ in different attention maps, quantified using the $\rho$-sparse metric $P_{\text{sparse}}(L, \rho)$ with different $\rho \in \{0.05, 0.02, 0.01\}$, as defined in Theorem 1. We estimate $P_{\text{sparse}}(L, \rho)$ using the observed sparsity frequency in the 100 examples. Additional results are provided in Figure 7.

## C.5. Implementation Details on Optimization Efficiency in Figure 3(a)

We train the GPT2-S and OPT-125M models on the SlimPajama dataset using a distributed setup consisting of 8 A800 GPUs. The training configuration includes a micro-batch size of 1, gradient accumulation over 8 steps, and a total of 1000 training steps. Integrating our proposed DGA into the model, we set the group size $m$ to 16 and the importance rate to 0.1. For GPT2-S, the learning rate is set to $2 \times 10^{-4}$, with a warmup period of 20 steps. For OPT-125M, the learning rate is set to $8 \times 10^{-5}$, with a warmup period of 50 steps and a weight decay coefficient of 0.1. Additionally, the AdamW optimizer is employed with the hyperparameters $\beta_1 = 0.9$ and $\beta_2 = 0.95$.

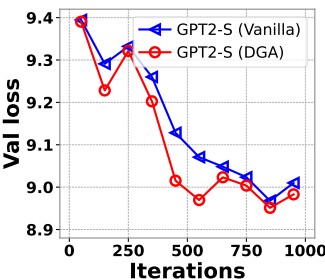

*Figure 4.* Validation losses of our DGA-LLM and vanilla ones on GPT2-S, where the models are trained with a 1K context length on SlimPajama.

# D. Details on Causing Mask in Group-Oriented Attention

---

**Algorithm 2** Constructing causing mask $\mathbf{M}$ for attention.

---

**input** Matrices $\mathbf{Q}, \mathbf{K}, \mathbf{V} \in \mathbb{R}^{L \times d}$, group size $m$, *focal tokens* $\mathcal{T}^{\text{foc}}$ and *non-focal tokens* $\mathcal{T}^{\text{non}}$.
**output** Causing mask $\mathbf{M} \in \mathbb{R}^{L \times (r+k+m)}$.
  1: Initialize a lower triangular matrix $\mathbf{M}^0 \in \mathbb{R}^{L \times L}$ with ones.
  2: Partition $\mathcal{T}^{\text{non}}$ into $k$ groups $\{G_i\}_{i=1}^k$, each of size $m$. // Note that non-divisible tokens are classified as focal tokens.
  3: Construct 0-1 mask $\mathbf{M}^n \in \mathbb{R}^{L \times k}$ for aggregated tokens by setting $\mathbf{M}_{i,j}^n = 1$ if $\|\mathbf{M}_{i,G_j}\|_0 > 0$.
  4: Construct 0-1 mask $\mathbf{M}^c \in \mathbb{R}^{L \times m}$ for complemented tokens:
$$\mathbf{M}_i^c = \Phi_{i,[m(z-1)+1:mz]} \in \mathbb{R}^m \text{ if } i \in [\min G_z, \max G_z],$$
    where $\Phi = \mathbf{M}_{\mathcal{T}^{\text{non}}}^0 - \phi^m(\mathbf{M}^n)$, $\phi^m(\mathbf{M}^n)$ repeats each column of the matrix $\mathbf{M}^n$ $m$ times.
  5: Final mask is $\mathbf{M} = [\mathbf{M}_{\mathcal{T}^{\text{foc}}}^0, \mathbf{M}^n, \mathbf{M}^c] \in \mathbb{R}^{L \times (r+k+m)}$.

---

## E. Analysis on Reduced Complexity and Key-value Cache

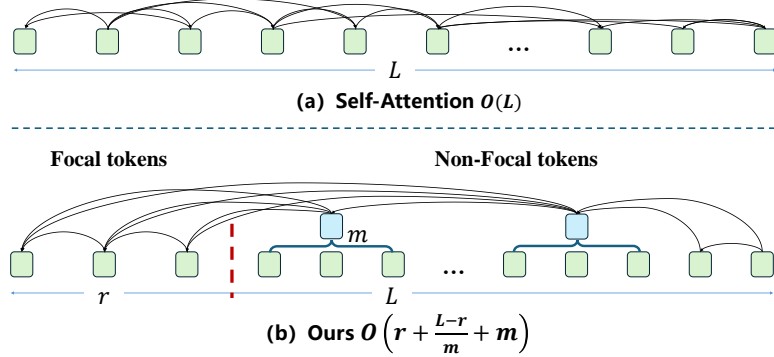

*Figure 5.* Comparison of the computational complexity during inference between our DGA-LLM and vanilla self-attention.

Our DGA-LLM enhances efficiency by adaptively grouping and aggregating redundant non-focal tokens, thereby reducing computational complexity and key-value cache memory storage compared to vanilla self-attention. Based on these benefits, our DGA-LLM achieves substantial improvements in terms of running speed and memory usage efficiency.

**Computational Complexity Optimization.** Our DGA-LLM exhibits varying computational complexities across training and inference phases. For the training phases, DGA-LLM achieves a computational complexity of $O(Lr + L\frac{L-r}{m} + Lm)$, representing a substantial improvement over the vanilla full self-attention mechanism, which has a complexity of $O(L^2)$. Specifically, the computation primarily involves three types of tokens: (1) focal tokens, contributing a complexity of $(Lr)$; (2) aggregated non-focal tokens, contributing $(L\frac{L-r}{m})$; (3) complement tokens, contributing $(Lm)$. Considering the group size $m$ and the number of focal tokens $r$ as constant, the final computational complexity can degenerate as $O(L^2/m)$.

As shown in Figure 5, during the inference phase, where the generation process involves next-token prediction, the computational complexity of generating a token is reduced to $O(r + \frac{L-r}{m} + m)$. This is significantly lower with large $m$ compared to the full self-attention mechanism, which has a complexity of $O(L)$.

**Key-Value (KV) Cache Reduction.** The KV cache is a widely used mechanism in self-attention-based Transformers to enhance the efficiency of attention computations by leveraging the model's auto-regressive nature to store and reuse key-value pairs (*i.e.*, $\mathbf{K}$ and $\mathbf{V}$). However, its memory consumption scales linearly with sequence length, making it a primary contributor to the overall memory footprint during inference. The increasing memory demand not only imposes substantial resource requirements but also potentially degrades inference speed due to the heightened frequency of I/O operations.

The full self-attention mechanism has a storage complexity of $O(L)$, whereas our proposed DGA-LLM significantly reduces it to $O(r + \frac{L-r}{m} + m)$. Specifically, instead of storing all original tokens at $O(L)$ complexity, we retain key and value matrices for three distinct token types: focal tokens with a complexity of $O(r)$, aggregated non-focal tokens at $O(\frac{L-r}{m})$, and complement tokens at $O(m)$.

## F. Ablation Studies

**Effect of group Size $m$.** In this experiment, we investigate the impact of group size $m$ on model performance and latency. As shown in Table 4, when varying $m$ from 4 to 64, we observe only a slight degradation in model performance, with perplexity (PPL) increasing from 3.20 to 3.87 and accuracy decreasing from 70.17% to 65.00%. However, computational efficiency shows a notable improvement, with latency significantly decreasing from 189.80 ms to 162.02 ms. Thus, we set $m = 16$ to strike a balance between computational efficiency and model performance as the default training setting.

*Table 4.* Effect of group size $m$.

| $m$ | 4 | 8 | 16 | 32 | 64 |
|---|---|---|---|---|---|
| PPL $\downarrow$ | 3.20 | 3.42 | 3.59 | 3.77 | 3.87 |
| Accuracy $\uparrow$ (%) | 70.17 | 68.33 | 67.00 | 65.61 | 65.00 |
| Latency $\downarrow$ (ms) | 189.80 | 189.10 | 172.12 | 169.40 | 162.02 |

**Effect of focal token identification.** In this experiment, we investigate the effect of different focal token identification strategies on model performance. From Table 5, compared to the randomly selected strategy, our Top-K approach achieves an improvement in accuracy from 66.91% to 67.00%. At the same time, perplexity (PPL) shows a decrease from 3.60 to 3.59. These results suggest the effectiveness of focal token identification in DGA over the randomly selected strategy.

*Table 5.* Effect of focal token identification strategy.

| Strategy | Randomly Selected | Top-K (Ours) |
|---|---|---|
| PPL $\downarrow$ | 3.60 | 3.59 |
| Accuracy $\uparrow$ (%) | 66.91 | 67.00 |

**Effect of importance rate $\gamma$.** In this experiment, we investigate the effect of the importance rate $\gamma$ in our DGA. Table 6 demonstrates a significant positive correlation between $\gamma$ values and model performance. Specifically, as $\gamma$ increases from 0.1 to 0.9, the model accuracy improves substantially from 67.00% to 70.58%, while perplexity (PPL) consistently decreases from 3.59 to 3.20. However, the latency shows an increase as $\gamma$ increases, growing from 172.12 ms to 555.25 ms. A larger $\gamma$ indicates that more tokens are identified as focal tokens, thereby preserving a greater portion of the original input information in the self-attention computations. This preservation of information, while beneficial for accuracy, inevitably increases the computational overhead. Conversely, a smaller $\gamma$ leads to more tokens being grouped and aggregated, which enhances computational efficiency but at the expense of model performance. Striking a balance between computational efficiency and model effectiveness, we select $\gamma = 0.1$ as the default training setting.

*Table 6.* Effect of importance rate $\gamma$.

| $\gamma$ | 0.1 | 0.3 | 0.5 | 0.7 | 0.9 |
|---|---|---|---|---|---|
| PPL $\downarrow$ | 3.59 | 3.43 | 3.27 | 3.22 | 3.20 |
| Accuracy $\uparrow$ (%) | 67.00 | 68.25 | 69.67 | 70.29 | 70.58 |
| Latency $\downarrow$ (ms) | 172.12 | 224.85 | 386.36 | 465.35 | 555.25 |

**Effect of complementary tokens.** Table 7 shows that removing complementary tokens significantly degrades performance on tasks like Multi-Doc QA from 3.58 to 2.37 and Code from 53.45 to 48.00, indicating complementary tokens are critical for context modeling. They slightly increase latency from 24.9 ms to 28.8 ms, but still $2.4 \sim 3.5\times$ faster than vanilla self-attention (Table 2: $69.70 \sim 102.22$ ms).

*Table 7.* Effect of complementary tokens on LongBench-E (Bai et al., 2024), where the 95% text length quantile is 31K. "ITL" denotes inter-token latency (Chitty-Venkata et al., 2024), which measures the time delay between generating consecutive tokens.

| Methods | Single Doc. QA | Multi Doc. QA | Summar. | FS Learning | Synthetic | Code | Avg. ↑ | ITL (ms) ↓ |
|---|---|---|---|---|---|---|---|---|
| w/o comple. tokens | 6.43 | 2.37 | 8.47 | 53.69 | 3.04 | 48.00 | 20.33 | 24.9 |
| DGA-LLM (Ours) | 3.61 | 3.58 | 6.81 | 57.90 | 1.47 | 53.45 | 21.14 | 28.8 |

## G. Discussions on DGA-LLM

**Hyperparameters selection**. Experimental analysis in Tables 4 and 7 reveals that smaller group sizes ($m$) and higher importance rates ($\gamma$) improve performance at the cost of increased latency. Based on these, we recommend the following practical guidelines for parameter selection: For resource-constrained scenarios, larger $m$ and lower $\gamma$ balance acceptable performance with reduced complexity. Performance-critical applications (*e.g.*, medical diagnosis) benefit from minimal $m$ and maximal $\gamma$ to preserve accuracy, whereas latency-sensitive tasks (*e.g.*, real-time systems) require moderate $m$ and lower $\gamma$ for responsiveness. An adaptive framework to automatically optimize ($m$, $\gamma$) based on application-specific accuracy-latency trade-offs remains a promising direction for future work.

**Potential application.** While our current work focuses on long-text modeling, the proposed DGA mechanism is inherently task-agnostic and could generalize to other long-sequence domains (*e.g.*, video/audio) where redundancy exists in sequential tokens and adaptive token importance assessment is critical. For instance, 1) Video processing: Temporal sequences in videos often exhibit localized redundancy (*e.g.*, static backgrounds or repetitive motions). DGA could dynamically group less informative frames while preserving critical temporal segments. 2) Audio processing: Long audio signals contain silent or redundant segments. DGA's importance scoring could prioritize phonetically rich regions, enabling efficient compression.

## H. More Experimental Results

### H.1. More Results on Redundancy in Self-attention

In this section, we present additional visualizations of the sparsity in vanilla attention weights. Figure 6 illustrates the distributions of attention weights across four randomly sampled long contexts from SlimPajama. Figure 7 depicts the distributions of $P_{\text{sparse}}(L, \rho)$ for $\rho \in \{0.05, 0.02, 0.01\}$. The results reveal that attention weights grow increasingly sparse with longer contexts, with only a small subset of tokens playing a significant role in predictions, aligning with the findings in Theorem 1 and the results in Section 6.2.

## I. Future Directions

While our proposed Dynamic Group Attention (DGA) demonstrates significant improvements in computational efficiency and robustness for long-context modeling, several avenues remain for further exploration. First, extending DGA to multi-modal settings, such as vision-language models or audio-text models, could unlock new possibilities for efficient cross-modal reasoning. For instance, integrating DGA with architectures like CLIP (Radford et al., 2021) or Flamingo (Alayrac et al., 2022) could enable more efficient processing of long video or audio sequences while maintaining high performance. Additionally, exploring the application of DGA in low-resource environments, such as edge devices or mobile platforms, could further enhance its practicality. Techniques like quantization (Frantar et al., 2022), distillation (Gu et al., 2024), or hardware-aware optimization (Artetxe et al., 2022) could be combined with DGA to reduce memory and computational requirements, making it suitable for real-time applications in resource-constrained settings.

Second, the theoretical foundations of DGA could be further refined to better understand its limitations and potential improvements. For example, a deeper analysis of the trade-offs between group size $m$ and model performance could provide insights into optimal hyperparameter settings for different tasks and datasets. Moreover, investigating the impact of DGA on downstream tasks, such as few-shot learning (Brown et al., 2020), transfer learning (Kenton & Toutanova, 2019), or continual learning (Ke et al., 2023), could reveal its broader applicability. Finally, exploring adaptive mechanisms for dynamically adjusting the group size $m$ or the importance threshold $\gamma$ during training and inference could further enhance the flexibility and efficiency of DGA. These directions not only address the limitations of the current approach but also open up new opportunities for advancing long-context modeling in both research and practical applications.

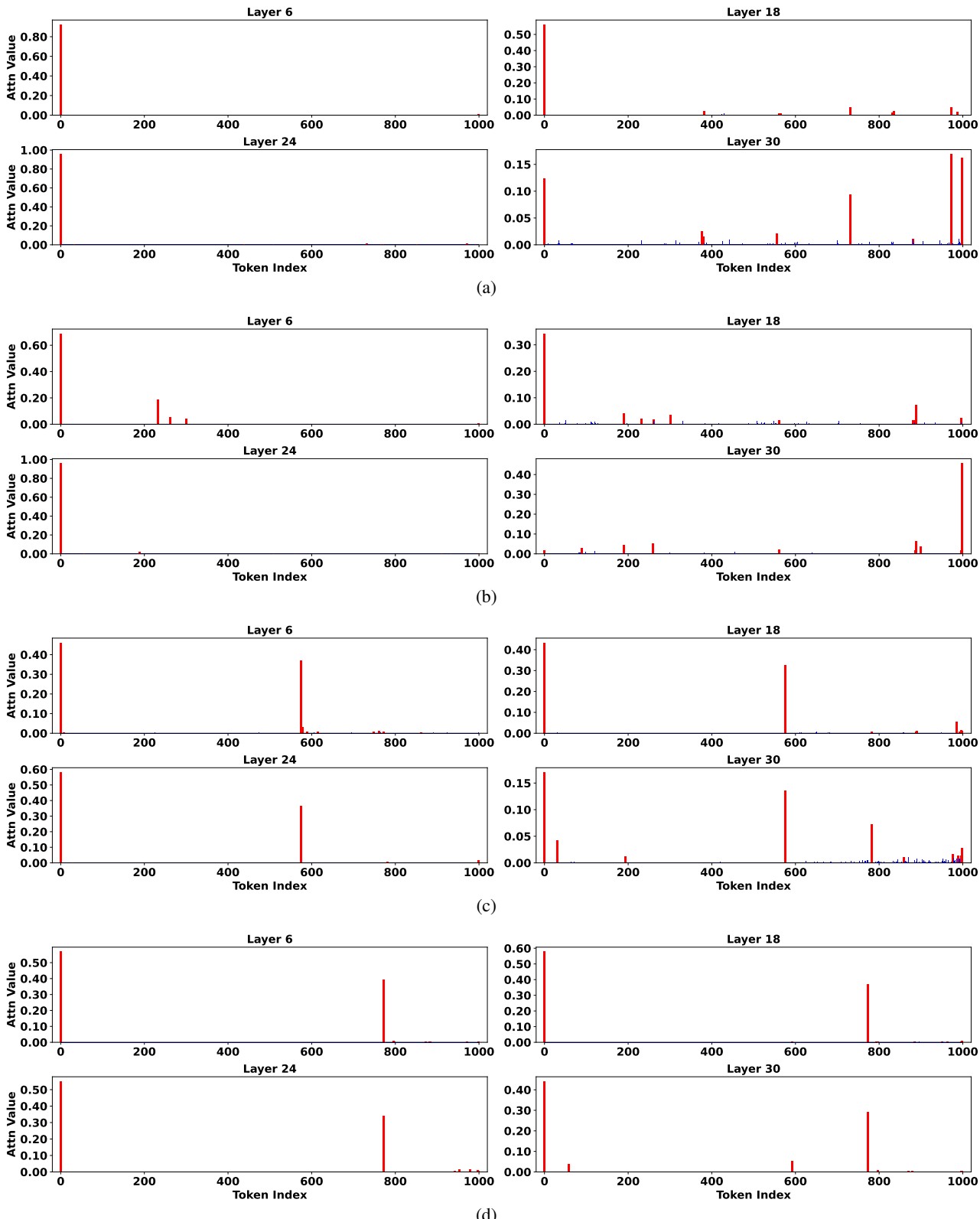

*Figure 6.* More visualizations on distribution of attention weights over Llama2-7B on 4 random examples from SlimPajama.

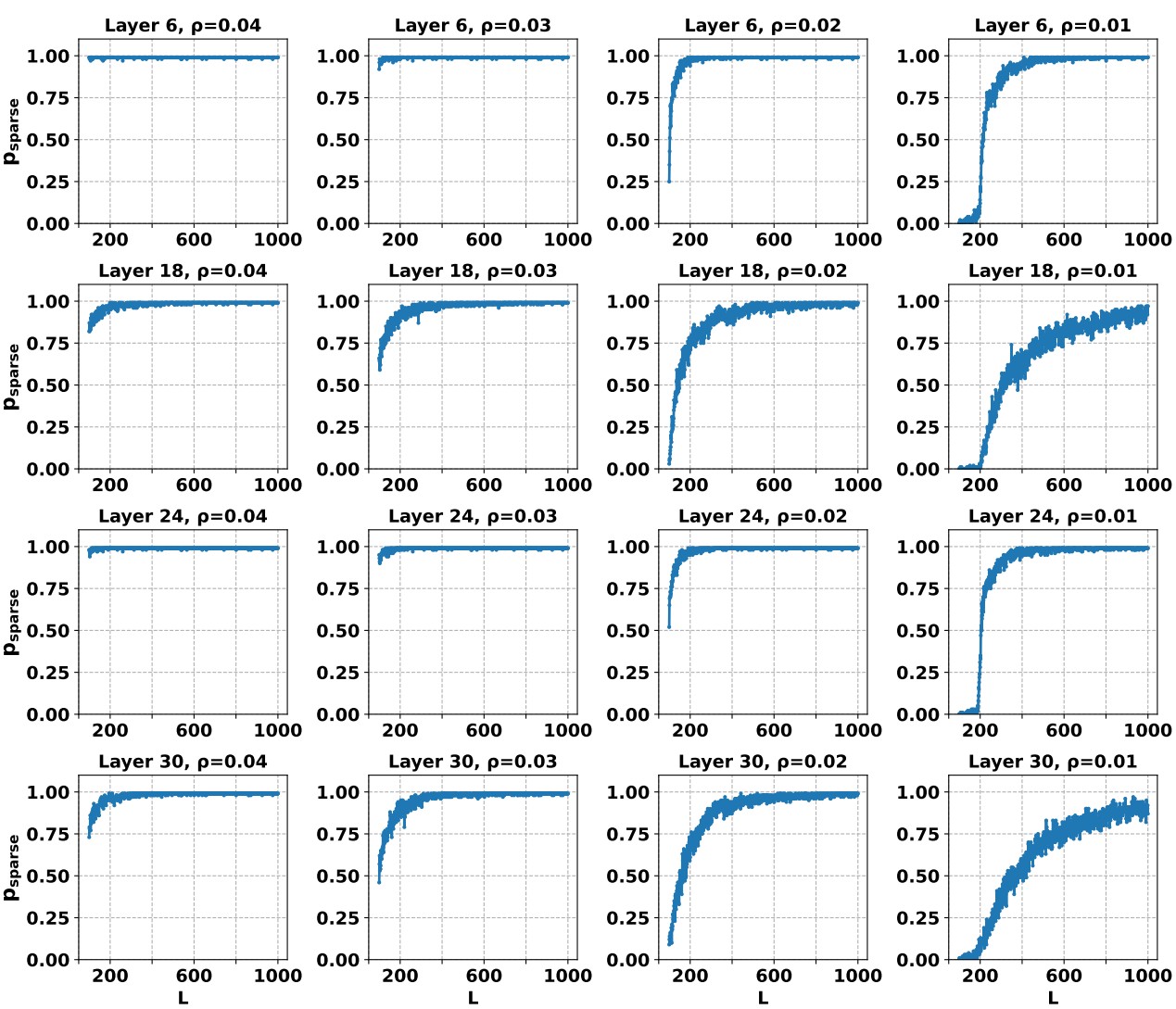

*Figure 7.* More visualizations on the sparsity of the attention weights.

