# OpenReview forum: "Curse of High Dimensionality Issue in Transformer for Long Context Modeling"
_ICML.cc/2025/Conference — ICML 2025 poster_

### Official Review · Reviewer_VyKb · 2025-03-03

**Overall Recommendation:** 4

**Summary:**

This paper explores the challenge of the curse of dimensionality in Transformer architectures for long-context modeling, with a particular focus on redundant attention computations. To address this issue, the authors introduce a novel approach called Dynamic Group Attention (DGA), which minimizes redundant computations by dynamically grouping and aggregating less critical tokens while preserving essential token interactions. They redefine traditional probabilistic sequence modeling as a supervised learning task and conduct a theoretical analysis of attention sparsity in Transformers, showing that only a small subset of tokens has a significant impact on predictions. Furthermore, they frame attention optimization as a linear coding problem and develop the DGA mechanism to dynamically manage token grouping and aggregation. Experimental results indicate that DGA effectively lowers computational costs while maintaining strong performance.

**Claims And Evidence:**

The claims made in the submission are supported by clear and convincing evidence.

**Essential References Not Discussed:**

The authors clearly discussed key methodologies in the field of efficient attention.

**Experimental Designs Or Analyses:**

The experimental designs and analyses in the paper are well-structured and robust.

* The authors compare the proposed method against a diverse set of baseline approaches, including MInference and StreamLLM, using LongBench-E and EM scores. The results demonstrate significant reductions in computational costs and inter-token latency while maintaining competitive performance across various long-context tasks.

My questions are:

* Can the proposed method be integrated with FlashAttention optimization strategies to achieve further acceleration? Additionally, does it support KV-Cache techniques?

* Do the baseline methods used for comparison incorporate FlashAttention or KV-Cache when evaluating latency?

**Methods And Evaluation Criteria:**

The proposed methods and evaluation criteria are well-suited to the problem, but I still have some concerns:

* While the DGA mechanism effectively mitigates redundant computations, its implementation and understanding are relatively complex. The dynamic grouping and aggregation strategy requires precise control, which may increase implementation difficulty and debugging costs.

* The DGA mechanism depends on several hyperparameters, such as the group size $m$ and the importance rate $\gamma$, which significantly influence both model performance and computational efficiency. Although the paper examines their impact experimentally, determining the optimal values in real-world applications remains challenging. A more in-depth discussion on hyperparameter selection would be beneficial.

**Other Comments Or Suggestions:**

Some formulas in the appendix are missing periods or commas, such as Equation 36, 42.

**Other Strengths And Weaknesses:**

* The proposed Dynamic Group Attention (DGA) mechanism introduces a novel strategy for optimizing attention mechanisms. By dynamically grouping and aggregating less important tokens, it effectively reduces redundant computations while preserving essential token interactions. This approach successfully lowers computational costs without compromising model performance, demonstrating strong practical value.

* The authors conduct extensive experiments across multiple long-context modeling tasks, including the LongBench-E benchmark. The results show that DGA not only reduces computational costs but also maintains or even outperforms existing methods in terms of performance. Notably, in long-text generation tasks involving extensive context lengths (e.g., 16K), DGA significantly reduces generation latency, underscoring its efficiency and suitability for long-context applications.

**Questions For Authors:**

The method in this paper is effective for long sequence text modeling. However, it remains unclear whether the proposed DGA mechanism can be applied to other long sequence processing tasks or architectures like video and audio processing. Without extra experiments, it would be great if the authors could discuss this potential application.

**Relation To Broader Scientific Literature:**

The authors clearly identify the issue of computational redundancy in Transformer-based long-context modeling and substantiate its presence through both theoretical analysis and empirical validation, establishing a strong foundation for further optimization.

Beyond introducing a novel method, the authors conduct an in-depth theoretical analysis of attention weight sparsity, reformulate the attention optimization problem as a linear coding issue, and propose a group coding strategy within this framework. This theoretical exploration offers a fresh perspective on understanding redundancy in Transformers.

While the paper primarily focuses on long-context modeling, the flexibility and efficiency of the Dynamic Group Attention (DGA) mechanism suggest its potential applicability to other tasks involving long-sequence processing, such as video and audio analysis. This highlights the broad applicability of the proposed method.

**Theoretical Claims:**

I have thoroughly reviewed the theoretical claims presented in the manuscript, including the proposed theorems and their corresponding proofs. The authors provide well-structured and insightful theoretical analyses that contribute valuable perspectives on the problem. This paper offers a rigorous and comprehensive understanding of redundancy in transformer-based long-context modeling, effectively identifying redundant tokens, providing probabilistic insights into attention sparsity, and introducing a robust optimization strategy through group coding. These theoretical contributions serve as a strong foundation for developing more efficient and effective attention mechanisms, as exemplified by the proposed Dynamic Group Attention mechanism.

---

> ### Author Rebuttal · Authors · 2025-03-31
>
> We really appreciate the reviewer's kind words and detailed suggestions. Here are our responses:
>
> >Q1. "...**well-structured and insightful theoretical analyses**... **strong foundation**...", "The experimental designs...are **well-structured and robust**...", "...theoretical exploration offers a **fresh perspective** on understanding redundancy", "...**flexibility and efficiency**... **potential applicability to other tasks**...", "...successfully lowers computational costs **without compromising model performance**, demonstrating strong **practical value**".
>
> **A1.** We sincerely thank the reviewer for their insightful feedback. We appreciate the recognition of our theoretical rigor in analyzing Transformer redundancy and the robust experimental validation demonstrating DGA’s efficiency-performance balance. The highlighted flexibility and practical value of our approach motivate us to explore broader applications in future work.
>
>
> >Q2. DGA mitigates redundancy well, but its implementation and understanding are complex. The dynamic grouping and aggregation strategy needs precise control, increasing implementation difficulty and debugging costs.
>
> **A2**. Thanks for your insightful comments. To address this concern, we will add a PyTorch-style pseudocode in the revised manuscript. Our approach leverages standard PyTorch operations, including matrix multiplication, `topk` selection, and tensor slicing, all of which are widely used and well-optimized within modern deep learning frameworks. We believe that this additional clarification will help elucidate the implementation.
>
> >Q3. >Q3. DGA depends on hyperparameters like $m$, $\gamma$, affecting performance. The paper examines their impact, but finding optimal values in practice is hard. Deeper discussion on selection would help.
>
> **A3**. Experimental analysis in Tables 4 and 6 reveals that smaller group sizes ($m$) and higher importance rates ($\gamma$) improve performance at the cost of increased latency. Based on these, we recommend the following practical guidelines for parameter selection: For resource-constrained scenarios, larger $m$ and lower $\gamma$ balance acceptable performance with reduced complexity. Performance-critical applications (e.g., medical diagnosis) benefit from minimal $m$ and maximal $\gamma$ to preserve accuracy, whereas latency-sensitive tasks (e.g., real-time systems) require moderate $m$ and lower $\gamma$ for responsiveness.
>
> An adaptive framework to automatically optimize ($m$, $\gamma$) based on application-specific accuracy-latency trade-offs remains a promising direction for future work.
>
> >Q4. Can the proposed method be integrated with FlashAttention optimization strategies to achieve further acceleration? Additionally, does it support KV-Cache techniques?
>
> **A4**. Yes, the proposed DGA method is compatible with FlashAttention and KV-Cache optimizations. For KV-Cache, DGA retains only 0.32–2.51GB of KV states on context-length 4K-32K (Table I), a 78–98% reduction compared to vanilla self-attention.
>
> Table I: Comparison of KV-Cache (GB) with Vanilla self-attention.
> | Methods | 4K | 8K | 16K | 32K |
> |---|---|---|---|---|
> | Vanilla self-attention | 2 | 4 | 8 | 16 |
> | DGA (Ours) | 0.32 | 0.63 | 1.26 | 2.51 |
>
> >Q5. Do the baseline methods used for comparison incorporate FlashAttention or KV-Cache when evaluating latency?
>
> **A5**. No. FlashAttention is not used in any method (including ours) to ensure fair comparison, as StreamingLLM's official implementation lacks support. All compared methods (including baselines and our DAG) employed standard KV-Cache techniques.
>
> >Q6. Some formulas in the appendix are missing periods or commas, such as Equation 36, 42.
>
> **A6**. We will fix it.
>
> >Q7. The method works for long text modeling. But it's unclear if DGA can be applied to other long sequence tasks (e.g., video, audio processing). Without extra experiments, could the authors discuss this potential application?
>
> **A7**. We appreciate the reviewer’s insightful question regarding the broader applicability of DGA. While our current work focuses on long-text modeling, the proposed Dynamic Grouping Attention (DGA) mechanism is inherently task-agnostic and could generalize to other long-sequence domains (e.g., video/audio) where redundancy exists in sequential tokens and adaptive token importance assessment is critical. For instance:
> * **Video Processing**: Temporal sequences in videos often exhibit localized redundancy (e.g., static backgrounds or repetitive motions). DGA could dynamically group less informative frames while preserving critical temporal segments.
> * **Audio Processing**: Long audio signals contain silent or redundant segments. DGA’s importance scoring could prioritize phonetically rich regions, enabling efficient compression.
>
> We will include the above discussions in our revised paper.

---

### Official Review · Reviewer_Q8qd · 2025-03-07

**Overall Recommendation:** 4

**Summary:**

In this paper, the authors propose a novel approach called Dynamic Group Attention (DGA) to address the computational inefficiencies in long-context modeling for transformer-based large language models. DGA leverages a group coding strategy to dynamically aggregate less important tokens while preserving critical token interactions. The approach aims to reduce redundancy in attention computations without sacrificing model performance. Through theoretical analysis, the authors demonstrate the robustness of the group coding mechanism and its ability to improve learning efficiency. Extensive experiments on the LongBench-E benchmark validate the effectiveness of DGA, showing significant reductions in computational costs while maintaining competitive performance across various long-context tasks.

**Claims And Evidence:**

This work successfully demonstrates the potential of Dynamic Group Attention (DGA) in addressing computational inefficiencies in long-context modeling for transformer-based large language models. The paper's claims are supported by solid evidence from theoretical analysis and extensive experiments.

**Essential References Not Discussed:**

The paper addresses key references relevant to its contributions. However, there is an existing method that uses dynamic grouping to accelerate attention[A]. It would be better to provide a more detailed discussion on how the proposed method differs from this.
[A] Dynamic Group Transformer: A General Vision Transformer Backbone with Dynamic Group Attention. (IJCAI 2022)

**Experimental Designs Or Analyses:**

The experimental designs and analyses presented in the paper are sound and robust.
1.The paper compares the proposed Dynamic Group Attention (DGA) with several baseline methods, including standard self-attention and other attention optimization techniques, providing a comprehensive evaluation of the method's performance.
2.The authors incorporate a variety of experiments, including comparisons on the LongBench-E benchmark, performance testing across multiple tasks, and inference efficiency evaluations, offering a solid foundation for validating the effectiveness of DGA.
3.Ablation studies are conducted to assess the contribution of key components, such as the group coding strategy and dynamic token aggregation, further supporting the claims of the method's benefits.
My questions are:
    How are the values of the thresholds (e.g., $\rho$) chosen? Are these values specific to certain datasets, and can they generalize across different datasets?
    It would be beneficial to include an ablation study on the complementary tokens, specifically evaluating their impact on the model's performance, particularly in terms of efficiency and accuracy.

**Methods And Evaluation Criteria:**

The methods presented in the paper are well-suited for the challenges of long-context modeling in transformer-based large language models.
My questions are:
1.	From Figure 1, it appears that the grouping operation involves reordering the tokens. When handling longer sequences, could this reordering introduce significant additional overhead?
2.	The paper would benefit from further clarification of certain method details. Specifically, it is unclear whether the group window size is consistent between the prefill and decoding stages. Additionally, more information is needed on how token importance is determined during the decoding stage.

**Other Comments Or Suggestions:**

NA

**Other Strengths And Weaknesses:**

The proposed method relies heavily on the sparsity of the long-context tokens' importance. However, in tasks with relatively lower sparsity, such as summarization, the method's performance seems to be less effective. Are there any potential solutions or adjustments that could improve its performance in such scenarios?

**Questions For Authors:**

1.	Can you provide a detailed computational complexity analysis of the proposed sparse attention mechanism, specifically for both the prefill and decoding stages?

**Relation To Broader Scientific Literature:**

The key contributions of this paper are strongly rooted in the existing literature, addressing significant gaps in long-context modeling for transformer-based large language models.

**Theoretical Claims:**

The theoretical claims in this paper are highly solid and well-supported. The introduction of Dynamic Group Attention (DGA) and its group coding strategy represents a significant innovation in addressing the challenges of long-context modeling. The theoretical analysis clearly demonstrates how DGA reduces computational redundancy and improves efficiency by dynamically aggregating less important tokens. This is well-justified and aligns perfectly with the experimental results, further validating the effectiveness of the method. The paper's theoretical background provides a strong foundation for understanding the improvements in attention mechanisms and model performance. Overall, the theoretical contributions contribute greatly to the novelty and impact of the work.

---

> ### Author Rebuttal · Authors · 2025-03-31
>
> We thank the reviewer for the encouraging comments and detailed suggestions. Responses are below:
>
> >Q1. "...**solid evidence** from **theoretical analysis** and **extensive experiments**", "...**theoretical analysis clearly demonstrates** how DGA reduces computational redundancy...**well-justified and aligns perfectly with** the experimental results", "...comparison...**comprehensive evaluation**...", "...incorporate **a variety of experiments**...", "...addressing **significant gaps** in long-context modeling..."
>
> **A1.** We are deeply appreciative of your thoughtful comments. The recognition that DGA’s redundancy reduction theory aligns with experimental results shows our approach's rigor. We appreciate your focus on comprehensive evaluation, including baseline comparisons and ablation studies, inspiring us to further develop solutions in this domain.
>
> >Q2.  From Fig. 1, grouping seems to reorder tokens. Could this introduce significant overhead for longer sequences?
>
> **A2.** No. The reordering operation introduces **minimal overhead** even for long sequences. From Table I, the repositioning time ratio increases only marginally (12.5%→15.8%) as context length grows from 4K to 16K, demonstrating scalability. Thus, the benefits of dynamic grouping (e.g., 2.4–3.5× latency reduction) far outweigh this minor cost.
>
> Table I: Percentage of total attention computation time for reordering operation across different sequence lengths (Table 3 settings).
> |Context Length|4K|8K|16K|
> |-|-|-|-|
> |Ratio|12.5%|15.6%|15.8%|
>
> >Q3. The paper needs clarity on method details, e.g., group window size consistency prefill-decoding, and token importance determination in decoding.
>
> **A3.** In decoding, we use a slightly larger group size ($m' = 1.1m$, e.g., 16→18) with 10% slots for focal tokens. Focal tokens are selected via top-10% attention weights from the group’s last token when a group reaches $m'$, ensuring adaptive prioritization and aligning with training principles. We'll clarify in revisions.
>
> >Q4. How are threshold values (e.g., ρ) chosen? Are they dataset-specific and generalizable?
>
> **A4.** The threshold $ρ$ is chosen based on sequence length $L$ ($ρ \in (1/L,1]$), with smaller $ρ$ enforcing stronger sparsity. Table II shows $P_{sparse}$ has **nearly identical trends** across SlimPajama and WikiText2, confirming ρ generalizes across datasets. As $ρ=0.01$, $P_{sparse}$ increases sharply with $L$, proving sparsity strengthens with context length universally. So, **ρ is dataset-independent and adapts to $L$**.
>
> Table II: Estimations of $P_{sparse}(L,ρ=0.01)$ across lengths (Fig. 2 settings).
> |L|100|200|300|400|500|600|700|800|900|1000|
> |-|-|-|-|-|-|-|-|-|-|-|
> |WikiText2|0.14|0.68|0.86|0.93|0.94|0.99|1.00|1.00|0.99|1.00|
> |SlimPajama|0.09|0.65|0.86|0.95|0.97|0.97|0.99|1.00|1.00|0.99|
>
> >Q5. Ablation study on complementary tokens, assessing impact on model performance (efficiency & accuracy).
>
> **A5.** Our ablation study (Table III) shows complementary tokens are critical: removing them significantly degrades performance on tasks like Multi-Doc QA (3.58→2.37) and Code (53.45→48.00). They slightly increase latency (24.9ms→28.8ms), but are still **2.4–3.5× faster** than vanilla self-attention (Table 2: 69.70–102.22ms).
>
> Table III: Effect of complementary tokens (Table 1 settings).
> |Methods|Single Doc. QA|Multi Doc. QA|Summar.|FS learning|Synthetic|Code|Avg.|ITL (ms)|
> |-|-|-|-|-|-|-|-|-|
> |w/o comple. tokens|6.43|2.37|8.47|53.69|3.04|48.00|20.33|24.9|
> |DGA-LLM (Ours)|3.61|3.58|6.81|57.90|1.47|53.45|21.14|28.8|
>
> >Q6. The proposed method relies on long-context token importance sparsity. In lower-sparsity tasks like summarization, its performance seems less effective. Any potential solutions for such scenarios?
>
> **A6.** For lower-sparsity tasks (e.g., summarization), our method can adjust group size m for balance performance and efficiency. Reducing m from 16 to smaller values (e.g., m = 2) improves summarization performance (9.53→6.81) but increases ITL (53.0ms→28.8ms). We will explore dynamic m-adjustment in future work.
>
> Table IV: Comparisons of inference performance for different group sizes (m=2→m=16) on LongBench-E.
> |**Task**|**vanilla self-attention**|**2**|**4**|**8**|**16 (default)**|
> |:-:|:-:|:-:|:-:|:-:|:-:|
> |Single Doc. QA|6.43|5.68|4.93|3.04|3.61|
> |Multi. Doc. QA|2.37|5.40|4.43|4.74|3.58|
> |Summar.|13.65|9.53|8.36|7.30|6.81|
> |ITL(ms)|69.7|53.0|38.3|33.5|28.8|
>
> >Q7. Provide detailed computational complexity analysis of sparse attention mechanism for prefill and decoding stages.
>
> **A7.**  Thanks for the suggestion. Let $L$, $r$, and $m$ denote the context length, the number of focal tokens, and the group size.
> * **Prefilling stage**: Complexity is $O(Lr+L\frac{L-r}{m}+Lm)$, simplifying to $O(\frac{L^2}{m})$ for constants $m$ and $r$, significantly lower than vanilla self-attention’s $O(L^2)$.
> * **Decoding stage**:, Per-token complexity is $O(r+\frac{L-r}{m} + m)$, significantly lower than vanilla’s $O(L)$ for large $m$.

---

### Official Review · Reviewer_HvaA · 2025-03-07

**Overall Recommendation:** 4

**Summary:**

This paper addresses the computational inefficiency in Transformer-based models for long-context modeling caused by redundant attention computations. The authors reformulate probabilistic sequence modeling as a supervised learning task, providing a theoretical foundation for analyzing redundancy. Building on this, they propose a group coding strategy to aggregate less important tokens and introduce Dynamic Group Attention (DGA), which dynamically groups non-focal tokens while preserving critical interactions. Experimental results show that DGA reduces computational costs while maintaining competitive performance on long-context benchmarks.

**Claims And Evidence:**

The claims in the paper are supported by clear evidence.

**Essential References Not Discussed:**

The authors clearly discussed key methodologies in the field of efficient attention.

**Experimental Designs Or Analyses:**

The authors conducts comprehensive empirical results that validate the effectiveness of DGA. The experiments on LongBench-E and EM score show significant reductions in computational costs and inter-token latency while maintaining competitive performance on various long-context tasks. The significant latency reduction underscores its practical utility for real-world applications requiring efficient long-context processing. Besides, the authors compare the proposed method with a wide range of baseline methods, including MInference and StreamLLM. The comparative evaluation provides sufficient evidence to support the authors' claims. I still have some questions:
1. In Table 3, are all the methods tested with Flash Attention?
2. The method introduces complementary tokens to recover masked information during autoregressive generation. However, I would like to understand the impact of these complementary tokens on the model's inference performance，such as latency, and overall accuracy.

**Methods And Evaluation Criteria:**

The methods presented in the paper are well-motivated, but I still have some question:
1. Some statement in Section 5 is confusing. The authors first mention that they divide the tokens into two parts. However, in Eqn. 12, I found the tokens are segmented into three parts.
2. Some notations are hard to understand, such as $min G_i$ in line 315. I guess it denote the minimum index in the group index $G_i$, right?
3. In Eqn 17, the authors use a small set of query to estimate the attention weights. However, what if we only have one query token in the decoding stage?

**Other Comments Or Suggestions:**

1.	Some titles appear odd and awkward, such as “C.4. Implementation Details on Sparsity” and “C.5. Implementation Details on Optimization Efficiency.”
2.	What is the meaning of the subfigures a b c and d in Figure 5?

**Other Strengths And Weaknesses:**

1.	In Algorithm 1, can the proposed method conduct attention parallelly? I found the proposed method seems to calculate attention for different query Q_i separately in line 8 of the algorithm.

**Questions For Authors:**

n/a

**Relation To Broader Scientific Literature:**

The proposed method seems to reduce the redundancy of the attention, different from the existing method like MInference and StreamLLM that directly discard some tokens.

**Theoretical Claims:**

I have thoroughly reviewed the theoretical claims presented in the manuscript, including the proposed theorems and their corresponding proofs. The authors offer clear theoretical insights. They rigorously establish the sparsity of attention weights (e.g., Theorem 1 on sparsity bounds), showing that only a small subset of tokens significantly contributes to predictions. Subsequently, they link the optimization in attention with group coding and show its improved robustness (Theorem 2) and optimization efficiency (Theorem 3). These thorough analyses deepen the understanding of the redundancy in the attention, providing a foundation for the proposed method to address the computational inefficiency in transformers.

---

> ### Author Rebuttal · Authors · 2025-03-31
>
> We thank the reviewer for the encouraging comments and suggestions. Responses are below:
>
> >Q1. "...**clear theoretical insights**...**deepen the understanding** of the redundancy in the attention, providing a **foundation** for the proposed method...", "...**comprehensive empirical results**...The comparative evaluation provides **sufficient evidence** to support the authors' claims", "...**different from the existing method** like MInference and StreamLLM that directly discard some tokens".
>
> **A1.** We sincerely appreciate your thoughtful and encouraging feedback on our work. Your acknowledgment of the theoretical insights into attention redundancy, along with the robust empirical validation, is greatly appreciated. We are also grateful for highlighting how our method innovatively diverges from approaches like MInference and StreamLLM by avoiding token discarding. These comments underscore the significance of our contributions, motivating us to further advance research in this domain.
>
> >Q2. Some statements in Section 5 are confusing. The authors first mention that they divide the tokens into two parts. However, in Eqn. 12, I found the tokens are segmented into three parts.
>
> **A2.** We appreciate the reviewer’s careful observation and clarify the token partitioning strategy:
> * **Two-Part Token Partitioning**: Tokens are divided into **focal** and **non-focal** (redundant) groups based on their importance scores.
> * **Complementary Tokens for Autoregressive Integrity**: To address potential information loss caused by grouping (e.g., some tokens cannot access the group information due to the autoregressive nature), we introduce **complementary KV pairs** (third part in Eq. 12) to restore missing dependencies.
>
> >Q3. Some notations are hard to understand, such as $min G_i$ in line 315. I guess it denotes the minimum index in the group index $G_i$, right?
>
> **A3.** Yes. $min G_i$ is used to identify the earliest token position in the group. We will clarify these notations in the revised manuscript to improve readability.
>
> >Q4. In Eqn 17, the authors use a small set of queries to estimate the attention weights. However, what if we only have one query token in the decoding stage?
>
> **A4.** In the decoding stage with only one query token, our method adapts as follows:
> * **Initial Generation Phase (Token Count < m)**: When fewer than $m$ tokens (group size) are generated, we directly compute attention using standard full self-attention (without grouping) to ensure accurate context capture. This avoids instability in weight estimation with limited tokens.
> * **Subsequent Generation (Token Count ≥ m)**: Once $m$ tokens are generated, we leverage the **query of the last token** in the current group to compute weights *P* (Eqn. 14). This query implicitly encodes dependencies on prior tokens through its positional encoding, enabling reliable importance estimation.
>
> >Q5. In Table 3, are all the methods tested with Flash Attention?
>
> **A5.** No. To ensure fair comparisons, none of the methods in Table 3 use Flash Attention because StreamingLLM's official implementation lacks support for it. However, our approach is compatible with Flash Attention optimizations (e.g., block matrix operations), which could further enhance performance in future implementations.
>
> >Q6. The method introduces complementary tokens to recover masked information during autoregressive generation. However, I would like to understand the impact of these complementary tokens on the model's inference performance, such as latency and overall accuracy.
>
> **A6.** Our ablation study (Table I below) shows complementary tokens are critical: removing them significantly degrades performance (21.14→20.33) on Longbench-E by **3.8%↓**. While they incur a slight latency increase (24.9ms→28.8ms), this remains **2.4–3.5× faster** than vanilla self-attention (Table 2: 69.70–102.22 ms).
>
> Table I: Effect of complementary tokens, where we train LLaMA2-7B on 8K context-length texts over SlimPajama and test on Longbench-E.
> |Methods|Single Doc. QA|Multi Doc. QA|Summar.|FS learning|Synthetic|Code|Avg.|ITL (ms)|
> |-|-|-|-|-|-|-|-|-|
> |w/o comple. tokens|6.43|2.37|8.47|53.69|3.04|48.00|20.33|24.9|
> |DGA-LLM (Ours)|3.61|3.58|6.81|57.90|1.47|53.45|21.14|28.8|
>
> >Q7. In Algorithm 1, can the proposed method conduct attention parallelly? I found that the proposed method seems to calculate attention for different query Q_i separately in line 8 of the algorithm.
>
> **A7.** Yes. While Algorithm 1 describes sequentially for clarity (e.g., line 8), our implementation uses batched matrix operations and GPU parallelism to process all queries in parallel.
>
> >Q8. Some titles appear odd and awkward, such as “C.4. Implementation Details on Sparsity” and “C.5. Implementation Details on Optimization Efficiency.” 10. What is the meaning of the subfigures a b c and d in Figure 5?
>
> **A8.** We will revise the appendix titles for clarity and add clear captions for the subfigures in Figure 5.

---

### Official Review · Reviewer_ta8R · 2025-03-19

**Overall Recommendation:** 3

**Summary:**

In this paper, the author proposes a new method for long context LLMs. It uses dynamic grouping to divide tokens into several groups,  the attention over coarse granularity of token groups achieves faster inference.

**Claims And Evidence:**

1. The LLM sparsity discovered in Sec 4 has actually already been revealed in several previous works (such as StreamingLLM).
2. The approach proposed in Sec 4 of performing grouping for tokens has already been proposed (see KVMerger "MODEL TELLS YOU WHERE TO MERGE: ADAPTIVE KV CACHE MERGING FOR LLMS ON LONG-CONTEXT TASKS"). Moreover, I believe the clustering-based approach used by KVMerger is simpler and more reasonable than the grouping method in this paper.
3. In Section 3, the author proposes using supervised learning to achieve long context feature extraction. However, it seems the author does not mention the supervised learning method in their method section (Sec 5).

**Essential References Not Discussed:**

Many related work are about grouping/clustering tokens in long-context LLM, such as KVMerger, PQCache: Product Quantization-based KVCache for Long Context LLM Inference, ClusterKV: Manipulating LLM KV Cache in Semantic Space for Recallable Compression

**Experimental Designs Or Analyses:**

Please see Methods And Evaluation Criteria

**Methods And Evaluation Criteria:**

1. The analysis in Figure 2, as well as the definition of ρ-sparse, is not as reasonable and clear as the analysis of attention sink in StreamingLLM (see Figure 2 of StreamingLLM).
2. Table 1 in the paper has issues: (1) The context window length setting is not specified by the author. (2) Since StreamingLLM performs token pruning, its inference efficiency should be much higher than the original model (LLaMA2-7B). However, in this table, StreamingLLM's speed (ITL) has decreased.
3. Similar issues appear in Table 3. StreamingLLM's inference efficiency should be almost unaffected by text length because it directly truncates text using a sliding window.

**Other Comments Or Suggestions:**

No

**Other Strengths And Weaknesses:**

The author has a very strange citation: they even included a citation for supervised learning (Hastie et al., 2009). This citation is clearly inappropriate.

**Questions For Authors:**

See Claims And Evidence and Methods And Evaluation Criteria

**Relation To Broader Scientific Literature:**

See Claims And Evidence.

**Theoretical Claims:**

No

---

> ### Author Rebuttal · Authors · 2025-03-31
>
> We thank the reviewer for the detailed comments. Responses are below:
>
> >Q1. The LLM sparsity discovered in Sec 4 has already been revealed in several previous works (e.g., StreamingLLM).
>
> **A1.**  We thank the reviewer for noting prior sparsity on self-attention weight like in StreamingLLM [r1]. Unlike empirical studies, we theoretically show why attention sparsity should happen and would be strengthened with context length (Theorem 1). Based on this, we developed Dynamic Group Attention (DGA), which adaptively aggregates tokens while preserving critical interactions. Theorems 2–3 further establish group coding's advantages in noise robustness and optimization efficiency. We highlight **two additional distinctions**:
> * **Static Sparsity vs. Dynamic Group Mechanism**. Methods like StreamingLLM often use static sparsity that prioritizes attention on initial/ final tokens, which may discard critical tokens; DGA dynamically groups tokens based on context-dependent importance, ensuring adaptation in dynamic scenarios.
> * **Perfomance Comparisons**. DGA outperforms StreamingLLM by 11.69% EM score (Table 2) with 1.28× speedup (Table I), highlighting both accuracy and efficiency gains.
>
> [r1] Efficient streaming language models with attention sinks. ICLR2024.
>
> >Q2.  The token grouping in Sec 4 has been proposed by KVMerger [r2]. Its clustering-based method seems simpler and more reasonable.
>
> **A2.** Our method differs from KVMerger [r2] in **objective**, **methodology** and **experiments**:
> * **Distinct Objectives**: KVMerger focuses on KV cache compression via Gaussian-kernel-based Key clustering but ignores attention computation redundancy; DGA targets self-attention acceleration in long-context tasks by dynamic grouping, thus **reducing both computation and memory consumption**.
> * **Methodological Differences**: KVMerger uses static Key clustering; DGA groups tokens by context-aware importance, adapting to input variations.
> * **Empirical Validation**: KVMerger lacks self-attention acceleration results; Our Table 3 shows **2.4× speedup** at 16K length with minimal accuracy loss.
>
> We will include this in our revised paper.
>
> [r2] Model Tells You Where to Merge: Adaptive KV Cache Merging for LLMs on Long-Context Tasks. Arxiv 2024.
>
> >Q3. It seems the author does not mention the supervised learning method in their method section (Sec 5).
>
> **A3.** Sec. 3 reformulates long-context modeling as a supervised learning task (Eq. 2), separating **relevant** (critical for predictions) and **irrelevant** (redundant for context) tokens, motivating **theoretical analysis of sparsity** (Theorem 1–3 in Sec. 4). This **inspires the design of DGA** in Sec. 5, where token relevance (Eq. 4) derived from supervised learning guides dynamic grouping in DGA ($s_i$ in Eq. 16), aggregating redundant tokens while preserving critical interactions. These three sections form a pipeline: **supervised learning identifies redundancy**→**theoretical analysis quantifies sparsity**→**DGA operationalizes efficient computation**. We will clarify this in our paper.
>
> >Q4. The Fig. 2 analysis and ρ-sparse definition seem less reasonable and clear than StreamingLLM's attention sink analysis (see Fig. 2 of StreamingLLM).
>
> **A4.** The ρ-sparse, i.e., $P_{sparse}(L,ρ)$ quantifies the probability of at least one attention weight exceeding $1/(Lρ)$, rigorously measuring how sparsity strengthens with context length $L$. It evaluates a model’s **inherent ability to prioritize critical tokens** in varying contexts, e.g., $P_{sparse}$ rises sharply for $ρ=0.01$ as $L$ grows (Fig. 2b).
>
> Crucially, we note the **different functionality**: StreamingLLM analyzes sparsity for **individual sequences** (e.g., attention maps), while our ρ-sparsity aggregates across **multiple sequences** (Sec. 6.2), offering a **model-level sparsity characterization**.
>
> >Q5. Issues in Tables 1 & 3: (1) Unclear context window lengths. (2) StreamingLLM is slower than vanilla despite pruning.
>
> **A5.** We address concerns:
> * **Context Window**: Models were trained on **8K context** and evaluated on **32K context** (covering 95% of LongBench-E sequences). ITL was at **16K context**.
> * **StreamingLLM Efficiency**:
>     * All methods in Tables 1 and 3 used MInference for consistent evaluation (with full KV caching overhead).
>     * With StreamingLLM’s official code (Table I), ITL stabilizes (36ms) but is higher than DGA-LLM (28ms) as it retains a fixed 4K-token cache, while **DGA dynamically reduces cache size** (e.g., 2576 tokens for 16K).
>
> Table I: ITL (ms) comparisons with official StreamingLLM.
> |Methods|4K|8K|16K|
> |-|-|-|-|
> |StreamingLLM|36.16|36.97|36.87|
> |**DGA (Ours)**|**26.26**|**26.87**|**28.79**|
>
> >Q6. Many related works are missing and the citation (Hastie et al., 2009) is clearly inappropriate.
>
> **A6.** We will carefully discuss these works according to the reviewer's suggestions.

---

### Decision · Program_Chairs · 2025-05-01

**Decision:**

Accept (poster)

**Comment:**

This paper presents a theoretically grounded and empirically validated method—Dynamic Group Attention (DGA)—to reduce redundant attention computation in long-context Transformers. The authors introduce a supervised learning-based formulation of attention sparsity and reformulate attention optimization as a group coding problem, supporting their claims with rigorous theorems and competitive performance on LongBench-E. While DGA shows clear improvements in latency and efficiency over baselines, some concerns remain regarding novelty relative to prior work (e.g., StreamingLLM, KVMerger) and clarity in method description. The authors provide detailed rebuttals addressing these issues, and while not all comparisons are compelling, the theoretical contributions and strong empirical results are valuable.